## PROCEEDINGS A

applied mathematics, complexity, statistical physics

thermodynamic limit, Kramers–Kronig relations, sum rules, Desai–Zwanzig model, Bonilla–Casado–Morilla model, order–disorder transitions

**Author for correspondence:**
Valerio Lucarini
e-mail: v.lucarini@reading.ac.uk

# Response theory and phase transitions for the thermodynamic limit of interacting identical systems

Valerio Lucarini[1,2], Grigorios A. Pavliotis[3] and Niccolò Zagli[1,2,3]

[1]Department of Mathematics and Statistics, University of Reading, and [2]Centre for the Mathematics of Planet Earth, University of Reading, Reading, UK
[3]Department of Mathematics, Imperial College London, London, UK

VL, 0000-0001-9392-1471

We study the response to perturbations in the thermodynamic limit of a network of coupled identical agents undergoing a stochastic evolution which, in general, describes non-equilibrium conditions. All systems are nudged towards the common centre of mass. We derive Kramers–Kronig relations and sum rules for the linear susceptibilities obtained through mean field Fokker–Planck equations and then propose corrections relevant for the macroscopic case, which incorporates in a self-consistent way the effect of the mutual interaction between the systems. Such an interaction creates a memory effect. We are able to derive conditions determining the occurrence of phase transitions specifically due to system-to-system interactions. Such phase transitions exist in the thermodynamic limit and are associated with the divergence of the linear response but are not accompanied by the divergence in the integrated autocorrelation time for a suitably defined observable. We clarify that such endogenous phase transitions are fundamentally different from other pathologies in the linear response that can be framed in the context of critical transitions. Finally, we show how our results can elucidate the properties of the Desai–Zwanzig model and of the Bonilla–Casado–Morillo model, which feature paradigmatic equilibrium and non-equilibrium phase transitions, respectively.

# 1. Introduction

Multi-agent systems are used routinely to model phenomena in the natural sciences, social sciences, and engineering. In addition to the standard applications of interacting particle systems to, e.g. plasma physics and stellar dynamics, phenomena such as cooperation [1], synchronization [2], systemic risk [3], consensus opinion formation [4,5] can be modelled using interacting multi-agent systems. Multi-agent systems are finding applications also in areas like management of natural hazards [6] and of climate change impacts [7]. We refer to [8] for a recent review on interacting multi-agent systems and their applications to the social sciences, and to [9] for a collections of articles showcasing their application in many different areas of science and technology. Additionally, multi-agent systems are also used as the basis for algorithms for sampling and optimization [10].

In this paper, we focus on a particular class of multi-agent systems, namely weakly interacting diffusions, for which the strength of the interaction between the agents is inversely proportional to the number of agents. Under the assumption of exchangeability, i.e. that the particles are identical, it is well known that one can pass to the limit as the number of agents goes to infinity, i.e. the *mean field limit*. In particular, in this limit the evolution of the empirical measure is described by a nonlinear, non-local Fokker–Planck equation, the *McKean–Vlasov Equation* [11,12]. We refer to [13] for a comprehensive review of the McKean–Vlasov equation from a theoretical physics viewpoint. The class of multi-agent models considered in this paper is sufficiently rich to include models for cooperation, systemic risk, synchronization, biophysics, and opinion formation.

An important feature of weakly interacting diffusions is that in the mean field (thermodynamic) limit they can exhibit phase transitions [1,4,14–16]. Phase transitions are characterized in terms of exchange of stability of non-unique stationary states for the McKean–Vlasov equation at the critical temperature/interaction strength.

In the case of equilibrium systems, such stationary states are associated with critical points of a suitably defined energy landscape. For example, for the Kuramoto model of nonlinear oscillators, at the critical noise strength the uniform distribution (on the torus) becomes unstable and stable localized stationary states emerge (phase-locking), leading to synchronization phase transition [17]. A complete theory of phase transitions for the McKean–Vlasov equation on the torus, that includes the Kuramoto model of synchronization, the Hegselmann–Krause model of opinion formation, the Keller–Segel model of chemotaxis. etc., is presented in [18], see also [19]. The effect of (infinitely) many local minima in the energy landscape on the structure of the bifurcation diagram was studied in [20]. Phase transitions for gradient system with local interactions were studied in [21,22]. Synchronization has been extensively discussed in the scientific literature, see [17,23–28].

## (a) Linear response theory

One of the main objectives of this paper is to investigate phase transitions for weakly interacting diffusions by looking at the response of the (infinite dimensional) mean field dynamics to weak external perturbations. We associate the nearing of a phase transition with the setting where a very small cause leads to very large effects, or, more technically, to the breakdown of linear response in the system, as described below.

Linear response theory provides a general framework for investigating the properties of physical systems [29]. Well-known applications of linear response theory include solid-state physics and optics [30] as well as plasma physics and stellar dynamics ([31], ch. 5). Furthermore, the range of systems for which linear response theory is relevant is very vast, see e.g. [32–36]. Recently, many new areas of applications of linear response theory are emerging across different disciplinary areas—see, e.g. a recent special issue [37]—and new formulations of the problem are being presented, where the conceptual separation between acting forcing and observed response is blurred [38]. In particular, recent applications of linear response theory include the prediction of climate response to forcings [39–44]. In modern terms, the goal is to define practical ways

to reconstruct the measure supported time-dependent pullback attractor [45] of the climate by studying the response to perturbations of a suitably defined reference climate state [46].

The mathematical theory of linear response for deterministic systems was developed by Ruelle in the context of Axiom A chaotic systems [47,48]. He provided explicit response formulae and showed that, in the case of dissipative systems, the classical fluctuation–dissipation theorem does not hold, and, as a result, natural and forced fluctuations are intimately different [49]. Ruelle's results have then been re-examined through a more a functional analytic lens by studying the impacts of the perturbations to the dynamics on the transfer operator [50] and then extended to a more abstract mathematical framework [51–53]. The direct implementation of Ruelle's formulae is extremely challenging, because of the radically different behaviour of the system along the stable and unstable manifold [54], which is related to the insightful *tout court* criticism of linear response theory by Van Kampen [55], so that alternative strategies have been devised [56,57]. Very promising progresses have been recently obtained in the direction of using directly Ruelle's formulae thanks to adjoint and shadowing methods [58–60].

Linear response theory and fluctuation–dissipation theorems have long been studied in detail for diffusion processes ([61], [62, ch. 7], [63, ch. 9]), and, more recently, rigorous results have been obtained in this direction [64,65]. An interesting link between response theory for deterministic and stochastic systems has been proposed in [66]. The results presented in [64,65] can be applied to the McKean–Vlasov equation *in the absence of phase transitions* to justify rigorously linear response theory and to establish fluctuation–dissipation results. See also [67] for formal calculations. This is not surprising, since it is well known that, in the absence of phase transitions, fluctuations around the mean field limit are Gaussian and can be described in terms of an appropriate stochastic heat equation [1,68].

## (b) Critical transitions versus phase transitions

Critical transitions appear when the spectral gap of the transfer operator [51] of the unperturbed system becomes vanishingly small, as a result of the Ruelle–Pollicott poles [69,70] touching the real axis. Since there is a one-to-one correspondence between the radius of expansion of linear response theory and the spectral gap of the transfer operator [51,71], near critical transitions the linear response breaks down and one finds rough dependence of the system properties on its parameters [72,73]. Systems undergoing critical transitions appear often in the natural and social sciences [74] and a lot of effort has been put in the development of early warning signals for critical transitions [75–77]. Early warning signals include an increase in variance and correlation time as the system approaches the transition point.

In the deterministic case, at the critical transition the reference state loses stability and the system ends up in a possibly very different metastable state. Indeed, the presence of critical transitions is closely related to the existence of regimes of multi-stability [78,79]. Transition points for finite-dimensional stochastic systems correspond to points where the topological structure of the *unique* invariant measure changes ([80], [63, Sect. 5.4]). Contrary to this, more than one invariant measure can exist in the mean field (thermodynamic) limit, e.g. in equilibrium systems, when the free energy is not convex [18]. The loss of uniqueness of the stationary state at the critical temperature/noise strength corresponds to a phase transition.

Phase transitions are usually defined by (a) identifying an order parameter and (b) verifying that in the thermodynamic limit, for some value of the parameter of the system, the properties of such an order parameter undergo a sudden change. It should be emphasized, however, that, for the mean field dynamics it is not always possible to identify an order parameter. The way we define phase transitions in this work comes from a somewhat complementary viewpoint, which aims at clarifying analogies and differences with respect to the case of critical transitions.

Sornette and collaborators have devoted efforts at separating the effects of endogeneous versus exogenous processes in determining the dynamics of a complex system and, especially in defining the conditions conducive to crises [81], and proposed multiple applications in the natural (e.g. [82]) as well as the social (e.g. [83]) sciences. The existence of a relationship between the

response of the system to exogeneous perturbations and the decorrelation due to endogenous dynamics is interpreted as resulting from a fluctuation–dissipation relation-like properties. Finally, Sornette and collaborators have also emphasized the importance of memory effects especially in the context of endogenous dynamics [84,85]. While our viewpoint and methods are different from theirs, what we pursued here shares similar goals and delves into closely related concepts.

## (c) This paper: goals and main results

*The main objective of this paper is to perform a systematic study of linear response theory for mean field partial differential equations (PDEs) exhibiting phase transitions.* Indeed, it has been shown that, for nonlinear oscillators coupled linearly with their mean, the so-called Desai–Zwanzig model [86], the fluctuations at the phase transition point are not Gaussian [1], see also [19] for related results for a variant of the Kuramoto model (the Haken–Kelso–Bunz model). Indeed, the fluctuations are persistent, non-Gaussian in time, with an amplitude described by a nonlinear stochastic differential equation, and associated with a longer timescale [1]. At the transition point, the standard form of linear response theory breaks down [14]. More general analyses performed using ideas from linear response theory of how a system of coupled maps performs a transition to a coherent state in the thermodynamic limit can be found in [87,88].

Here, we consider a network of $N$ identical and coupled $M$-dimensional systems whose evolution is described by a Langevin equation. We then study the response to perturbations in the limit of $N \to \infty$. We investigate the conditions determining the breakdown of the linear response and separate two possible scenarios. One scenario pertains to the closure of the spectral gap of the transfer operator of the mean field equations, and can be dealt with through the classical theory of critical transitions. A second scenario of breakdown of the linear response results from the coupling among the $N$ systems and is inherently associated with the thermodynamic limit. We focus on the second scenario of breakdown of the linear response, which we interpret as corresponding to a phase transition. The main results of this paper can be summarized as follows:

— the derivation of linear response formulae for the thermodynamic limit of a network of coupled identical systems and of Kramers–Kronig relations and sum rules for the related susceptibilities;
— the statement of conditions leading to phase transitions as opposed to the classical scenario of critical transitions;
— the explicit derivation of the corrections to the standard Kramers–Kronig relations and sum rules occurring at the phase transition;
— the clarification, through the use of functional analytical arguments, of why one does not expect divergence of the integrated autocorrelation time of suitable observables in the case of phase transitions, whereas the opposite holds in the case of critical transitions;
— the re-examination, also through numerical simulations, of classical results on phase transitions in the Desai–Zwanzig model [86] and in the Bonilla–Casado–Morillo model [89].

The rest of the paper is organized as follows. In §2, we introduce our model and present the linear response formulae for the mean field equations as well as for the renormalized macroscopic case. In §3, we discuss the properties of the frequency-dependent susceptibility, present the Kramers–Kronig relations connecting their real and imaginary parts, and find explicit sum rules. In §4 we discuss under which conditions the response diverges, and clarify the fundamental difference between the case of critical transitions and the case of phase transitions, which can take place only in the thermodynamic limit. Section 5 is dedicated to finding results that specifically apply to the case of gradient systems, corresponding to reversible Markovian dynamics. In §6 we re-examine the case of phase transitions for the Desai–Zwanzig and Bonilla–Casado–Morilla models, which are relevant for the case of equilibrium and non-equilibrium

dynamics, respectively. Finally, in §7, we present our conclusions and provide perspectives for future investigations.

## 2. Linear response formulae: mean field and macroscopic results

We consider a network of $N$ exchangeable interacting $M$-dimensional systems whose dynamics is described by the following stochastic differential equations:

$$dx_i^k = F_{i,\alpha}(\mathbf{x}^k)dt - \frac{\theta}{N}\sum_{l=1}^{N}\partial_{x_i^k}U\left(\mathbf{x}^k - \mathbf{x}^l\right)dt + \sigma s_{ij}(\mathbf{x})dW_j, \quad k=1,\ldots,N\ i=1,\ldots,M \quad (2.1)$$

where $\mathbf{F}_\alpha$ is a smooth vector field, possibly depending on a parameter $\alpha$. Additionally, $dW_i$, $i=1,\ldots,N$ are independent Brownian motions (the Ito convention is used); $s_{ij}$ is the volatility matrix, and the parameter $\sigma > 0$ controls the intensity of the stochastic forcing. Additionally, the $N$ systems undergo an all-to-all coupling through the Laplacian matrix given by the derivative of the potential $U(\mathbf{y}) = |\mathbf{y}|^2/2$. We emphasize the fact that the linear response theory calculations presented below are valid for arbitrary choices of the interaction potential. We choose to present our results for the case of quadratic interactions since in this case the order parameter is known; furthermore, the stationary state(s) are known and are parametrized by the order parameter [1]. The coefficient $\theta$ modulates the intensity of such a coupling, which attempts at synchronizing all systems by nudging them to the centre of mass $1/N\sum_{k=1}^{N}\mathbf{x}^k$. If $\theta = 0$, the $N$ systems are decoupled. We remark that the theory of synchronization says that for this choice of the coupling, if $d\mathbf{x}/dt = \mathbf{F}_\alpha(\mathbf{x})$ has a unique attractor and is chaotic with $\lambda_1 > 0$ being the largest Lyapunov exponent, the $N$ nodes undergo perfect synchronization for any $N \geq 2$ in the absence of noise ($\sigma = 0$) if $\theta > \lambda_1$ [17,28,90,91].

If $\mathbf{F}_\alpha(\mathbf{y}) = -\nabla V_\alpha(\mathbf{y})$, we interpret $V_\alpha$ as the confining potential [63]. In some cases, equation (2.1) describes an equilibrium statistical mechanical system, in particular if $\mathbf{F}_\alpha = -\nabla V_\alpha(\mathbf{y})$ and $s_{ij}$ is proportional to the identity. More generally, equilibrium conditions are realized when the drift term—the deterministic component on the right-hand side of equation (2.1)—is proportional to the gradient of a function defined according to the Riemannian metric given by the diffusion matrix $C_{ij} = s_{ik}s_{jk}$ [92].

We now consider the empirical measure $\rho^{(N)}$, which is defined as $\rho^{(N)} = 1/N\sum_{k=1}^{N}\delta_{\mathbf{x}^k(t)}$. Following [11,18,93], we investigate the thermodynamic limit of the system above. As $N \to \infty$, we can use martingale techniques [1,11,93,94] to show that the one-particle density converges to some measure $\rho(\mathbf{x},t)$ satisfying the following McKean–Vlasov equation, which is a nonlinear and non-local Fokker–Planck equation :

$$\frac{\partial\rho(\mathbf{x},t)}{\partial t} = -\nabla \cdot [\rho(\mathbf{x},t)(\mathbf{F}_\alpha(\mathbf{x}) - \theta\nabla U \star \rho)] + \frac{\sigma^2}{2}\tilde{\Delta}\rho(\mathbf{x},t)$$

$$= -\nabla \cdot [\rho(\mathbf{x},t)(\mathbf{F}_\alpha(\mathbf{x}) + \theta(\langle\mathbf{x}\rangle(t) - \mathbf{x}))] + \frac{\sigma^2}{2}\tilde{\Delta}\rho(\mathbf{x},t)$$

$$:= L_{\alpha,\theta}^0(\rho(\mathbf{x},t)) + \theta\Lambda_\theta(\{\rho(\mathbf{x},t)\}), \quad (2.2)$$

where we have separated the linear operator $L_{\alpha,\theta}^0$ and the nonlinear operator $\Lambda_\theta(\{\rho(\mathbf{x},t)\}) = \theta\nabla \cdot (\rho(\mathbf{x},t)\langle\mathbf{x}\rangle(t))$, with $\langle\mathbf{x}\rangle(t) = \int d^M\mathbf{y}\rho(\mathbf{y},t)$ and $\star$ denotes the convolution. Additionally, we have that $\tilde{\Delta}$ is a linear diffusion operator such that $\tilde{\Delta}\rho(\mathbf{x},t) = \sum_{i=1}^{M}\sum_{j=1}^{M}\partial_{x_i}\partial_{x_j}(C_{ij}(\mathbf{x})\rho(\mathbf{x},t))$, which coincides with the standard M-dimensional Laplacian ($\tilde{\Delta} = \Delta$) if the diffusion matrix $C_{ij}$ is the identity matrix. If $\sigma = 0$, we are considering a nonlinear Liouville equation. We assume that, if $\sigma > 0$, equation (2.1) describes a hypoelliptic diffusion process, so that $\rho_{\alpha,\theta}^{(0)}(\mathbf{x})$ is smooth [63, ch. 6]. In what follows, we refer to the case $\sigma > 0$. Conditions detailing the well-posedness of this problem can be found in [95].

Let us define $\rho_{\alpha,\theta}^{(0)}(\mathbf{x})$ as a reference invariant measure of the system such that $L_{\alpha,\theta}^0(\rho_{\alpha,\theta}^{(0)}(\mathbf{x})) + \theta \Lambda_\theta(\{\rho_{\alpha,\theta}^{(0)}(\mathbf{x})\}) = 0$. Since we are considering a system with an infinite number of particles, such an invariant measure needs not be unique [1,18,96,97]. Specifically, if $s_{ij}$ is proportional to the identity and $\mathbf{F}_\alpha(\mathbf{y}) = -\nabla V_\alpha(\mathbf{y})$ and $V_\alpha(\mathbf{y})$ is not convex, thus allowing for more than one local minimum, for a given value of $\theta$ the system undergoes a phase transition for sufficiently weak noise; see discussion in §5.

We remark that the invariant measure depends on the values of $\alpha$ and $\theta$, and, in particular, $\langle \mathbf{x} \rangle(t) = \langle \mathbf{x} \rangle_{\alpha,\theta}^{(0)} = \langle \mathbf{x} \rangle_0$ is a constant vector, where in the last identity we have dropped the lower indices to simplify the notation. As a result, we have that:

$$M_{\alpha,\theta,\langle\mathbf{x}\rangle_0}^0\left(\rho_{\alpha,\theta}^{(0)}(\mathbf{x})\right) = -\nabla \cdot \left(\rho_{\alpha,\theta}^{(0)}(\mathbf{x})\left(\mathbf{F}(\mathbf{x}) + \theta\left(\langle\mathbf{x}\rangle_0 - \mathbf{x}\right)\right)\right) + \frac{\sigma^2}{2}\tilde{\Delta}\rho_{\alpha,\theta}^{(0)}(\mathbf{x}) = 0 \tag{2.3}$$

so that the invariant measure $\rho_{\alpha,\theta}^{(0)}(\mathbf{x})$ is the eigenvector with vanishing eigenvalue of the linear operator $M_{\alpha,\theta,\langle\mathbf{x}\rangle_0}^0$.

Taking inspiration from [98,99], we now study the impact of perturbations on the invariant measure $\rho_{\alpha,\theta}^{(0)}(\mathbf{x})$. We follow and extend the results presented in [67]. We modify the right-hand side of equation (2.2) by setting $\mathbf{F}_\alpha(\mathbf{x}) \to \mathbf{F}_\alpha(\mathbf{x}) + \epsilon \mathbf{X}(\mathbf{x})T(t)$ and we study the linear response of the system in terms of the density $\rho(\mathbf{x}, t)$. We then write $\rho(\mathbf{x}, t) = \rho_{\alpha,\theta}^{(0)}(\mathbf{x}) + \epsilon\rho_{\alpha,\theta}^{(1)}(\mathbf{x}, t) + o(\epsilon^2)$ and obtain the following equation up to order $\epsilon$:

$$\begin{aligned}
\frac{\partial \rho_{\alpha,\theta}^{(1)}(\mathbf{x}, t)}{\partial t} &= M_{\alpha,\theta,\langle\mathbf{x}\rangle_0}^0(\rho_{\alpha,\theta}^{(1)}(\mathbf{x}, t)) - T(t)\nabla \cdot \left(\rho_{\alpha,\theta}^{(0)}(\mathbf{x})\mathbf{X}(\mathbf{x})\right) \\
&\quad - \theta\nabla \cdot \left(\rho_{\alpha,\theta}^{(0)}(\mathbf{x})\int d^M\mathbf{y}\rho_{\alpha,\theta}^{(1)}(\mathbf{y}, t)\mathbf{y}\right) \\
&= \tilde{M}_{\alpha,\theta,\langle\mathbf{x}\rangle_0}^0(\rho_{\alpha,\theta}^{(1)}(\mathbf{x}, t)) - T(t)\nabla \cdot \left(\rho_{\alpha,\theta}^{(0)}(\mathbf{x})\mathbf{X}(\mathbf{x})\right)
\end{aligned} \tag{2.4}$$

We remark that the linear operator $\tilde{M}_{\alpha,\theta,\langle\mathbf{x}\rangle_0}^0$ acting on $\rho_{\alpha,\theta}^{(1)}(\mathbf{x}, t)$ on the right-hand side of the previous equation is not the operator whose zero eigenvector is the unperturbed invariant measure. The correction proportional to $\theta$ emerges as a result of the nonlinearity of the McKean–Vlasov equation. We will discuss the operator $\tilde{M}_{\alpha,\theta,\langle\mathbf{x}\rangle_0}^0$ in §2a.

One then derives:

$$\begin{aligned}
\rho_{\alpha,\theta}^{(1)}(\mathbf{x}, t) &= \int_{-\infty}^t ds \exp\left(M_{\alpha,\theta,\langle\mathbf{x}\rangle_0}^0(t-s)\right)\left[-T(s)\nabla \cdot \left(\rho_{\alpha,\theta}^{(0)}(\mathbf{x})\mathbf{X}(\mathbf{x})\right)\right] \\
&\quad + \int_{-\infty}^t ds \exp\left(M_{\alpha,\theta,\langle\mathbf{x}\rangle_0}^0(t-s)\right)\left[-\theta\nabla \cdot \left(\rho_{\alpha,\theta}^{(0)}(\mathbf{x})\int d^M\mathbf{y}\rho_{\alpha,\theta}^{(1)}(\mathbf{y}, s)\mathbf{y}\right)\right]
\end{aligned} \tag{2.5}$$

We now evaluate the response of the observable $x_i$. This is sufficient for our purposes, since we know that, for this model, the order parameter (in the mean field limit) is the mean position $\langle \mathbf{x} \rangle$ (magnetization). By definition, we have that

$$\langle x_i \rangle(t) = \langle x_i \rangle_0 + \langle x_i \rangle_1(t) + O(\epsilon^2),$$

where we have defined $\langle \Phi \rangle_0 = \int d^M\mathbf{y}\rho_{\alpha,\theta}^{(0)}(\mathbf{y})\Phi(\mathbf{y})$ and $\langle \Phi \rangle_1(t) = \int d^M\mathbf{y}\rho_{\alpha,\theta}^{(1)}(\mathbf{y}, t)\Phi(\mathbf{y})$ for a generic observable $\Phi$. We obtain:

$$\begin{aligned}
\langle x_i \rangle_1(t) &= \int_{-\infty}^t ds \int d^M\mathbf{y}\rho_{\alpha,\theta}^{(0)}(\mathbf{y})\mathbf{X}(\mathbf{y})T(s) \cdot \nabla \exp\left(M_{\alpha,\theta,\langle\mathbf{x}\rangle_0}^{0,+}(t-s)\right)y_i \\
&\quad + \theta\int_{-\infty}^t ds \int d^M\mathbf{y}\rho_{\alpha,\theta}^{(0)}(\mathbf{y})\langle\mathbf{x}\rangle_1(s) \cdot \nabla \exp\left(M_{\alpha,\theta,\langle\mathbf{x}\rangle_0}^{0,+}(t-s)\right)y_i,
\end{aligned} \tag{2.6}$$

where we have defined the following operator:

$$M_{\alpha,\theta,\langle\mathbf{x}\rangle_0}^{0,+} = \mathbf{F}(\mathbf{x}) \cdot \nabla + \theta\left(\langle\mathbf{x}\rangle_0 - \mathbf{x}\right) \cdot \nabla + \frac{\sigma^2}{2}\tilde{\Delta}^+ \tag{2.7}$$

where $O^+$ is the adjoint of $O$. Following [67], we can interpret this as the Koopman operator for the unperturbed dynamics; see later discussion. We can rewrite the previous expression as:

$$\langle x_i \rangle_1(t) = \int_{-\infty}^{\infty} ds\, T(s) G_{i,\alpha,\theta}(t-s) + \sum_{k=1}^{M} \int_{-\infty}^{\infty} ds\, \langle x_k \rangle_1(s) Y_{\{i,k\},\alpha,\theta}(t-s) \tag{2.8}$$

where

$$G_{i,\alpha,\theta}(\tau) = \Theta(\tau) \int d^M \mathbf{y} \left( \rho_{\alpha,\theta}^{(0)}(\mathbf{y}) \mathbf{X}(\mathbf{y}) \right) \cdot \nabla \exp \left( M_{\alpha,\theta,\langle \mathbf{x} \rangle_0}^{0,+}(\tau) \right) y_i \tag{2.9}$$

and

$$Y_{\{i,k\},\alpha,\theta}(\tau) = \theta \Theta(\tau) \int d^M \mathbf{y}\, \rho_{\alpha,\theta}^{(0)}(\mathbf{y}) \partial_{y_k} \exp \left( M_{\alpha,\theta,\langle \mathbf{x} \rangle_0}^{0,+}(\tau) \right) y_i, \tag{2.10}$$

where the Green function is causal. Note also that if $\mathbf{X}(\mathbf{x}) = \hat{\mathbf{v}}_k$, where $\hat{\mathbf{v}}_k$ is the unit vector in the $k$th direction, then $G_{i,\alpha,\theta}(\tau) = Y_{\{i,k\},\alpha,\theta}(\tau)/\theta$.

Notwithstanding the Markovianity of the dynamics, the second term on the right-hand side of equation (2.8) describes a memory effect in the response of the observable $\mathbf{x}$. Such a term emerges in the thermodynamic limit and effectively imposes a condition of self-consistency between forcing and response; see different yet related results obtained by Sornette and collaborators [81,84,85].

If $\sigma > 0$ the invariant measure is smooth, so that we can perform an integration by parts of the previous expressions and derive the following Green functions:

$$G_{i,\alpha,\theta}(\tau) = -\Theta(\tau) \int d^M \mathbf{y}\, \rho_{\alpha,\theta}^{(0)}(\mathbf{y}) \frac{\nabla \cdot \left( \rho_{\alpha,\theta}^{(0)}(\mathbf{y}) \mathbf{X}(\mathbf{y}) \right)}{\rho_{\alpha,\theta}^{(0)}(\mathbf{y})} \exp \left( M_{\alpha,\theta,\langle \mathbf{x} \rangle_0}^{0,+} \tau \right) y_i \tag{2.11}$$

and

$$Y_{\{i,k\},\alpha,\theta}(\tau) = -\theta \Theta(\tau) \int d^M \mathbf{y}\, \rho_{\alpha,\theta}^{(0)}(\mathbf{y}) \partial_{y_k} \log \left( \rho_{\alpha,\theta}^{(0)}(\mathbf{y}) \right) \exp \left( M_{\alpha,\theta,\langle \mathbf{x} \rangle_0}^{0,+} \tau \right) y_i, \tag{2.12}$$

where the Green functions are written as correlation functions times a Heaviside distribution enforcing causality.

We remark that we can, at least formally, write:

$$\exp \left( M_{\alpha,\theta,\langle \mathbf{x} \rangle_0}^{0,+} t \right) = \Pi_0 + \sum_{j=1}^{\infty} \exp \left( t \lambda_j \right) \Pi_j + \mathcal{R}(t), \tag{2.13}$$

where $\{\lambda_j\}_{j=1}^{\infty}$ are the eigenvalues (point-spectrum) of $M_{\alpha,\theta,\langle \mathbf{x} \rangle_0}^{0,+}$ and $\Pi_j$ is the spectral projector onto the eigenspace spanned by the eigenfunction $\psi_j$, and in particular, $\Pi_0$ projects on the invariant measure. Then, the operator $\mathcal{R}(t)$ is the residual operator associated with the essential spectrum. The norm of $\mathcal{R}(t)$ is controlled by the distance of essential spectrum from the imaginary axis.

We then have:

$$G_{i,\alpha,\theta}(\tau) = \Theta(\tau) \sum_{j=1}^{\infty} \langle \psi_j y_i \rangle_0 \langle \Phi_X \psi_j \rangle_0 \exp \left( \lambda_j t \right) + \mathcal{R}_{\Phi_X}(\tau) \tag{2.14}$$

and

$$Y_{\{i,k\},\alpha,\theta}(\tau) = \Theta(\tau) \sum_{j=1}^{\infty} \langle \psi_j y_i \rangle_0 \langle \Phi_k \psi_j \rangle_0 \exp \left( \lambda_j t \right) + \mathcal{R}_{\Phi_k}(\tau), \tag{2.15}$$

where $\Phi_X = -\nabla \cdot (\rho_{\alpha,\theta}^{(0)}(\mathbf{y}) \mathbf{X}(\mathbf{y}))/\rho_{\alpha,\theta}^{(0)}(\mathbf{y})$ and $\Phi_k = -\theta \partial_{y_k} \log(\rho_{\alpha,\theta}^{(0)}(\mathbf{y}))$. Note that the $j = 0$ term vanishes because the corresponding scalar product $\langle \Phi_X \psi_0 \rangle_0$ has nil value for any choice of the vector field $\mathbf{X}$.

We now apply the Fourier transform $\mathcal{F}$ to equation (2.8) and obtain:

$$P_{ij,\alpha,\theta}(\omega)\langle x_j\rangle_1(\omega) = \Gamma_{i,\alpha,\theta}(\omega)T(\omega) \quad P_{ij,\alpha,\theta}(\omega) = \delta_{ij} - \Upsilon_{\{i,j\},\alpha,\theta}(\omega) \tag{2.16}$$

where we have used a (standard) abuse of notation in defining the Fourier transform of $T(t)$ and $\langle x_j\rangle_1(t)$ and have defined

$$\Gamma_{i,\alpha,\theta}(\omega) = \mathcal{F}\{G_{i,\alpha,\theta}(\omega)\} = \sum_{j=1}^{\infty} \frac{\langle \psi_j y_i\rangle_0 \langle \Phi_X \psi_j\rangle_0}{i\omega + \lambda_j} + \mathcal{R}_{\Phi_X}(\omega) \tag{2.17}$$

and

$$\Upsilon_{\{i,k\},\alpha,\theta}(\omega) = \mathcal{F}\{Y_{\{i,k\},\alpha,\theta}(t)\} = \sum_{j=1}^{\infty} \frac{\langle \psi_j y_i\rangle_0 \langle \Phi_k \psi_j\rangle_0}{i\omega + \lambda_j} + \mathcal{R}_{\Phi_k}(\omega). \tag{2.18}$$

We remark that the susceptibilities given in equations (2.17) and (2.18) are holomorphic in the upper complex $\omega$-plane if $\mathbf{Re}\{\lambda_j\} < 0$, $j = 1, \ldots, \infty$. Note that all susceptibilities, regardless of the observable considered, share the same poles located at $\omega_j = i\lambda_j$, $j = 1, \ldots, \infty$. Additionally, if $\omega_j$ is a pole, so is also $-\omega_j^*$ (and, correspondingly, $\lambda_j$ comes together with $\lambda_j^*$).

By introducing the inverse matrix $\Pi_{\alpha,\theta} = P_{\alpha,\theta}^{-1}$, we obtain from equation (2.16) our final result:

$$\langle x_i\rangle_1(\omega) = \Pi_{ij,\alpha,\theta}(\omega)\Gamma_{i,\alpha,\theta}(\omega)T(\omega) = \tilde{\Gamma}_{i,\alpha,\theta}(\omega)T(\omega) \tag{2.19}$$

where:

$$\tilde{\Gamma}_{i,\alpha,\theta}(\omega) = \Pi_{ij,\alpha,\theta}(\omega)\Gamma_{j,\alpha,\theta}(\omega). \tag{2.20}$$

The previous expression generalizes previous findings presented in [87]. We will discuss below the invertibility properties of the matrix $P_{ij,\alpha,\theta}(\omega)$. If the coupling is absent, so that $\theta = 0$, we obtain the same result as in the case of a single particle $N = 1$ system: $\langle x_i\rangle_1(\omega) = \Gamma_{i,\alpha,\theta=0}(\omega)T(\omega)$. Additionally, we trivially get $\Gamma_{i,\alpha,\theta=0}(\omega) = \tilde{\Gamma}_{i,\alpha,\theta=0}(\omega)$. The effect of switching on the coupling and taking $\theta > 0$ is twofold in terms of response:

— First, the function $\Gamma_{i,\alpha,\theta}(\omega)$ is modified, because the unperturbed evolution operator $M^0_{\alpha,\theta,\langle\mathbf{x}\rangle_{0,\theta}}$ (see equation (2.4)) and the unperturbed invariant measure $\rho^{(0)}_{\alpha,\theta}(\mathbf{x})$ depend explicitly on $\theta$. Indeed, changes in the value of $\theta$ impact expectation values and correlation properties. From the definition of $M^0_{\alpha,\theta,\langle\mathbf{x}\rangle_0}$, we interpret $\Gamma_{i,\alpha,\theta}(\omega)$ as the mean field susceptibility.

— More importantly, the presence of a non-vanishing value of $\theta$ introduces a non-trivial correction with respect to the identity to the matrix $P_{ij,\alpha,\theta}(\omega)$. We can interpret the function $\tilde{\Gamma}_{i,\alpha,\theta}(\omega)$ as the macroscopic susceptibility, which takes fully into account, in a self-consistent way, the interaction between the systems. Equation (2.19) generalizes the frequency-dependent version of the well-known Clausius–Mossotti relation [30,100, 101], which connects the macroscopic polarizability of a material and the microscopic polarizability of its elementary components.

The integration by parts used for deriving equations (2.11) and (2.12) from equations (2.9) and (2.10) amounts to deriving a variant of the fluctuation–dissipation relation [29,33], as the Green functions are written as the causal part of a time-lagged correlation of two observables as determined by unperturbed dynamics. In other terms, the poles $\omega_j$, $j = 1 \ldots, \infty$ of the susceptibilities above correspond to the Ruelle–Pollicott poles [69,70] of the unperturbed system, just as in the case of systems described by the standard Fokker–Planck equation [73,102]. This establishes a close connection between forced and free variability or, using a different terminology, between the properties of response to exogenous perturbations and endogenous dynamics [81].

## (a) Another expression for the macroscopic susceptibility

A somewhat unsatisfactory aspect of the previous derivation resides in the fact that we are dealing with the operator $\exp(M^{0,+}_{\alpha,\theta,\langle\mathbf{x}\rangle_0}t)$, which is associated with the mean field approximation. We can instead proceed from equation (2.5) using the operator $\exp(\tilde{M}^{0}_{\alpha,\theta,\langle\mathbf{x}\rangle_0}t)$ introduced above and derive directly the following results:

$$\rho^{(1)}_{\alpha,\theta}(\mathbf{x},t) = \int_{-\infty}^{t} \mathrm{d}s \exp\left(\tilde{M}^{0}_{\alpha,\theta,\langle\mathbf{x}\rangle_0}(t-s)\right)\left[-T(s)\nabla\cdot\left(\rho^{(0)}_{\alpha,\theta}(\mathbf{x})\mathbf{X}(\mathbf{x})\right)\right] \tag{2.21}$$

and

$$\langle x_i\rangle_1(t) = \int_{-\infty}^{t} \mathrm{d}s \int \mathrm{d}^M\mathbf{y}\,\rho^{(0)}_{\alpha,\theta}(\mathbf{y})\mathbf{X}(\mathbf{y})T(s)\cdot\nabla\exp\left(\tilde{M}^{0,+}_{\alpha,\theta,\langle\mathbf{x}\rangle_0}(t-s)\right)y_i T(s). \tag{2.22}$$

We can rewrite the previous expression as:

$$\langle x_i\rangle_1(t) = \int_{-\infty}^{\infty} \mathrm{d}s\,T(s)\tilde{G}_{i,\alpha,\theta}(t-s), \tag{2.23}$$

where the Fourier transform of:

$$\tilde{G}_{i,\alpha,\theta}(\tau) = \Theta(\tau)\int \mathrm{d}^M\mathbf{y}\left(\rho^{(0)}_{\alpha,\theta}(\mathbf{y})\mathbf{X}(\mathbf{y})\right)\cdot\nabla\exp\left(\tilde{M}^{0,+}_{\alpha,\theta,\langle\mathbf{x}\rangle_0}(\tau)\right)y_i$$

$$= -\Theta(\tau)\int \mathrm{d}^M\mathbf{y}\,\nabla\cdot\left(\rho^{(0)}_{\alpha,\theta}(\mathbf{y})\mathbf{X}(\mathbf{y})\right)\exp\left(\tilde{M}^{0,+}_{\alpha,\theta,\langle\mathbf{x}\rangle_0}(\tau)\right)y_i \tag{2.24}$$

is the macroscopic susceptibility introduced in equation (2.19). Note that $\tilde{M}^{0,+}_{\alpha,\theta,\langle\mathbf{x}\rangle_0}$ *cannot* be interpreted as the generator of time translation for smooth observables.

Clearly, the benefit of deriving the expression of $\tilde{\Gamma}_{i,\alpha,\theta}(\omega)$ as done in the previous section lies in the possibility of bypassing the space-integral operator included in the definition of $\tilde{M}^{0,+}_{\alpha,\theta,\langle\mathbf{x}\rangle_0}$. Similar to equation (2.13), we can write:

$$\exp\left(\tilde{M}^{0,+}_{\alpha,\theta,\langle\mathbf{x}\rangle_0}t\right) = \sum_{j=1}^{\infty}\exp\left(t\tilde{\lambda}_j\right)\tilde{\Pi}_j + \tilde{\mathcal{R}}(t), \tag{2.25}$$

where the corresponding symbols are used. We then have:

$$\tilde{\Gamma}_{i,\alpha,\theta}(\tau) = \Theta(\tau)\sum_{j=1}^{\infty}\langle\tilde{\psi}_j y_i\rangle_0\langle\Phi_\Gamma\tilde{\psi}_j\rangle_0\exp\left(\tilde{\lambda}_j t\right) + \tilde{\mathcal{R}}_{\Phi_X}(\tau). \tag{2.26}$$

We now apply the Fourier transform to equation (2.26) and obtain:

$$\tilde{\Gamma}_{i,\alpha,\theta}(\omega) = \sum_{j=1}^{\infty}\frac{\langle\psi_j y_i\rangle_0\langle\Phi_X\psi_j\rangle_0}{i\omega+\tilde{\lambda}_j} + \tilde{\mathcal{R}}_{\Phi_\Gamma}(\omega) \tag{2.27}$$

Comparing equations (2.27) and (2.20), it is clear that the poles $\tilde{\omega}_j$ of $\tilde{\Gamma}_{i,\alpha,\theta}(\omega)$ are those of $\Gamma_{i,\alpha,\theta}(\omega)$ plus those of the matrix $\Pi_{ij,\alpha,\theta}(\omega)$, see earlier comments by Dawson [1] for the case of the Desai–Zwanzig model [86] (see also §6a).

# 3. Dispersion relations far from criticalities

We assume that all $\lambda_j$, $j = 1,\ldots,\infty$ have negative real part. As discussed above, since $G^{(1)}_{j,\alpha,\theta}(\tau)$ is causal, the function $\Gamma^{(1)}_{j,\alpha,\theta}(\omega)$ is a well-behaved susceptibility function that is holomorphic in the upper complex $\omega$-plane ($\mathbf{Im}\{\omega\} \geq 0$).

Let us now consider the short-time behaviour $\tau \to 0^+$ of the response functions $G_{i,\alpha,\theta}(\tau)$. Using equations (2.7) and (2.9), we derive:

$$G_{i,\alpha,\theta}(\tau) = \Theta(\tau) \left( \langle X_i(\mathbf{x}) \rangle_0 + \left( \sum_{k=1}^{M} \langle X_k(\mathbf{x}) \partial_{x_k} F_i(\mathbf{x}) \rangle_0 - \theta \langle X_i(\mathbf{x}) \rangle_0 \right) \tau + o(\tau^2) \right). \tag{3.1}$$

As a result, the high-frequency behaviour of the susceptibility $\Gamma_{i,\alpha,\theta}(\omega)$ can be written as:

$$\Gamma_{i,\alpha,\theta}(\omega) = i \frac{\langle X_i(\mathbf{x}) \rangle_0}{\omega} - \frac{\sum_{k=1}^{M} \langle X_k(\mathbf{x}) \partial_{x_k} F_i(\mathbf{x}) \rangle_0 - \theta \langle X_i(\mathbf{x}) \rangle_0}{\omega^2} + o(\omega^2). \tag{3.2}$$

The causality of $G_{i,\alpha,\theta}(\tau)$ implies that, using an abuse of notation, $G_{i,\alpha,\theta}(\tau) = \Theta(\tau)G_{i,\alpha,\theta}(\tau)$. By performing the Fourier transform of both sides of this identity, we obtains the following identity $\Gamma_{i,\alpha,\theta}(\omega) = \frac{1}{2\pi}\Gamma_{i,\alpha,\theta}(\omega) \star \tilde{\Theta}(\omega))$, where $\star$ indicates the convolution product and $\tilde{\Theta}(\omega) = -i\mathbf{P}(1/\omega) + \pi\delta(\omega)$ is the Fourier transform of $\Theta(\tau)$, with $\mathbf{P}$ indicating the principal part. By separating the real (**Re**) and imaginary (**Im**) parts of $\Gamma_{i,\alpha,\theta}(\omega)$, the previous relation can be written as:

$$\mathbf{P} \int_{-\infty}^{\infty} d\nu \frac{\mathbf{Re}\{\Gamma_{i,\alpha,\theta}(\nu)\}}{\nu - \omega} = -\pi \mathbf{Im}\{\Gamma_{i,\alpha,\theta}(\omega)\} \tag{3.3}$$

and

$$\mathbf{P} \int_{-\infty}^{\infty} d\nu \frac{\mathbf{Im}\{\Gamma_{i,\alpha,\theta}(\nu)\}}{\nu - \omega} = \pi \mathbf{Re}\{\Gamma_{i,\alpha,\theta}(\omega)\}. \tag{3.4}$$

Since $\Gamma_{i,\alpha,\theta}(\tau)$ is a real function of real argument $\tau$, its Fourier transform obeys the following conditions: $\Gamma_{i,\alpha,\theta}(\omega) = (\Gamma_{i,\alpha,\theta}(-\omega^*))^*$. Hence, for real values of $\omega$ we have $\mathbf{Re}\{\Gamma_{i,\alpha,\theta}(\omega)\} = \mathbf{Re}\{\Gamma_{i,\alpha,\theta}(-\omega)\}$ and $\mathbf{Im}\{\Gamma_{i,\alpha,\theta}(\omega)\} = -\mathbf{Im}\{\Gamma_{i,\alpha,\theta}(-\omega)\}$. We derive an alternative form of the Kramers–Kronig relations [30]:

$$\mathbf{P} \int_0^{\infty} d\nu \frac{\mathbf{Re}\{\Gamma_{i,\alpha,\theta}(\nu)\}}{\nu^2 - \omega^2} = -\frac{\pi}{2\omega} \mathbf{Im}\{\Gamma_{i,\alpha,\theta}(\omega)\} \tag{3.5}$$

and

$$\mathbf{P} \int_0^{\infty} d\nu \frac{\nu \mathbf{Im}\{\Gamma_{i,\alpha,\theta}(\nu)\}}{\nu^2 - \omega^2} = \frac{\pi}{2} \mathbf{Re}\{\Gamma_{i,\alpha,\theta}(\omega)\}. \tag{3.6}$$

It is then possible to derive the following sum rules:

$$\int_0^{\infty} d\nu \mathbf{Re}\{\Gamma_{i,\alpha,\theta}(\nu)\} = \lim_{\omega \to \infty} \left( \frac{\pi}{2} \omega \mathbf{Im}\{\Gamma_{i,\alpha,\theta}(\omega)\} \right) = \frac{\pi}{2} \langle X_i(\mathbf{x}) \rangle_0 \tag{3.7}$$

and

$$\int_0^{\infty} d\nu \frac{\mathbf{Im}\{\Gamma_{i,\alpha,\theta}(\nu)\}}{\nu} = \lim_{\omega \to 0} \left( \frac{\pi}{2} \mathbf{Re}\{\Gamma_{i,\alpha,\theta}(\omega)\} \right) = \frac{\pi}{2} \tau_{G_i} G_{i,\alpha,\theta}(0^+), \tag{3.8}$$

where $\tau_{G_i} = \int_0^{\infty} dt G_{i,\alpha,\theta}(t)/G_{i,\alpha,\theta}(0^+)$, if $G_{i,\alpha,\theta}(0^+) \neq 0$ is a measure of the decorrelation of the system, see a related result in [103] on the Desai–Zwanzig model [86] discussed below. Note that $\mathbf{Im}\{\Gamma_{i,\alpha,\theta}(\omega)\}$ is an odd function of $\omega$. Additionally, if $\langle X_i(\mathbf{x}) \rangle_0 = 0$, so that the imaginary part of the susceptibility decreases asymptotically at least as fast as $\omega^{-3}$, the following additional sum rules holds:

$$\int_0^{\infty} d\nu \nu \mathbf{Im}\{\Gamma_{i,\alpha,\theta}(\nu)\} = \lim_{\omega \to \infty} \left( -\frac{\pi}{2} \omega^2 \mathbf{Re}\{\Gamma_{i,\alpha,\theta}(\omega)\} \right) = \frac{\pi}{2} \sum_{k=1}^{M} \langle X_k(\mathbf{x}) \partial_{x_k} F_i(\mathbf{x}) \rangle_0. \tag{3.9}$$

Let us now look at the asymptotic properties for large values of $\omega$ of the matrix $P_{ij,\alpha,\theta}(\omega)$. We proceed as above and consider the short time behaviour of $Y_{\{i,j\},\alpha,\theta}(\tau)$:

$$Y_{\{i,j\},\alpha,\theta}(\tau) = \Theta(\tau) \left( \delta_{ij}\theta + o(\tau) \right). \tag{3.10}$$

As a result, for large values of $\omega$, we have that

$$\Upsilon_{\{i,k\},\alpha,\theta}(\omega) = i \frac{\theta}{\omega} \delta_{i,k} + o(\omega^{-1}) \tag{3.11}$$

so that $P_{ij,\alpha,\theta}(\omega) = \delta_{ij}(1 - i(\theta/\omega)) + o(\omega^{-2})$ and $\Pi_{ij,\alpha,\theta}(\omega) = \delta_{ij}(1 + i(\theta/\omega)) + o(\omega^{-2})$, so that:

$$\tilde{\Gamma}_{i,\alpha,\theta}(\omega) = i\frac{\langle X_i(\mathbf{x})\rangle_0}{\omega} - \frac{\sum_{k=1}^{M}\langle X_k(\mathbf{x})\partial_{x_k}F_i(\mathbf{x})\rangle_0}{\omega^2} + o(\omega^2) \tag{3.12}$$

where we note a correction in the asymptotic behaviour with respect to the case of the mean field susceptibility given in equation (3.2). Nonetheless, if $P_{ij,\alpha,\theta}$ has full rank for all values of $\omega$ in the upper complex $\omega$-plane, the Kramers–Kronig relations (3.5) and (3.6) and the sum rules (3.7)–(3.9) apply also for the macroscopic susceptibilities $\tilde{\Gamma}_{i,\alpha,\theta}(\omega)$.

# 4. Criticalities

We remark again that the dispersion relations presented above apply for the mean field susceptibilities for values of $\alpha$ and $\theta$ such that (i) the real part of all the eigenvalues of $M_{\alpha,\theta,\langle\mathbf{x}\rangle_0}^{0,+}$ is negative; and for the macroscopic susceptibility if, additionally, (ii) the matrix $P_{ij,\alpha,\theta}$ is invertible and, additionally, has no zeros in the upper complex $\omega$-plane. Conditions (i) and (ii) correspond to the case where the real part of all the eigenvalues of $\tilde{M}_{\alpha,\theta,\langle\mathbf{x}\rangle_0}^{0,+}$ is negative.

The breakdown of condition (i) for, say, $(\alpha,\theta) = (\bar{\alpha},\bar{\theta})$ is due to the presence of a vanishing spectral gap for the operator $M_{\alpha,\theta,\langle\mathbf{x}\rangle_0}^{0,+}$, and, *a fortiori*, for the operator $\tilde{M}_{\alpha,\theta,\langle\mathbf{x}\rangle_0}^{0,+}$. In such a scenario, the functions $\Gamma_{i,\bar{\alpha},\bar{\theta}}(\omega)$ and $\tilde{\Gamma}_{i,\bar{\alpha},\bar{\theta}}(\omega)$ feature one or more poles in the real $\omega$-axis. In other terms, linear response blows up for forcings having non-vanishing spectral power $|T(\omega)|^2$ at the corresponding frequencies.

In this case, because of the link discussed above between the poles of the mean field susceptibilities and the Ruelle–Pollicott poles of the unperturbed system, the blow-up of the linear susceptibilities corresponds to an ultraslow decay of correlations leading to a singularity in the integrated decorrelation time. In other terms, in this case the results conform to the classic framework of the theory of critical transitions [46,72,73,76,104]. We remark that the presence of a divergence does not depend on the specific functional form of the perturbation field $\mathbf{X}$, while the properties of the response do depend in general from it.

The breakdown of condition (ii) for, say, $(\alpha,\theta) = (\tilde{\alpha},\tilde{\theta})$ is associated with the fact that the spectral gap of the operator $\tilde{M}_{\tilde{\alpha},\tilde{\theta},\langle\mathbf{x}\rangle_0}^{0,+}$ vanishes, while the spectral gap of the operator $M_{\tilde{\alpha},\tilde{\theta},\langle\mathbf{x}\rangle_0}^{0,+}$ remains finite. In this latter case, only the functions $\tilde{\Gamma}_{i,\tilde{\alpha},\tilde{\theta}}(\omega)$ have one or more poles for real values of $\omega$, whereas the functions $\Gamma_{i,\tilde{\alpha},\tilde{\theta}}(\omega)$ are holomorphic in the upper complex $\omega$-plane. We remark that the non-invertibility of the $P$ matrix depends on the presence of sufficiently strong coupling between the systems, which leads to them being coordinated, as discussed in detail in §6.

The nonlinearity of equation (2.2) emerges as a result of the thermodynamic limit $N \to \infty$. Therefore, we interpret the singularities in the linear response resulting from the breakdown of condition (ii) as being associated to a phase transition of the system, yet not a standard one. Indeed, the blow-up of the linear susceptibilities *does not* correspond to a blow-up of the integrated correlation time (see §6a).

## (a) Phase transitions

In what follows, we focus on the criticalities associated with condition (ii) only, which emerge *specifically* from effects that cannot be described using the mean field approximation.

Let's then assume that for some reference values for $\alpha = \alpha_0$ and $\theta = \theta_0$ the system is stable. This corresponds to the fact that the inverse Fourier transform of $\tilde{\Gamma}_{i,\alpha_0,\theta_0}(\omega)$, which defines a renormalized linear Green function that takes into account all the interactions among the identical systems, has only positive support. Correspondingly, the macroscopic susceptibilities $\tilde{\Gamma}_{i,\alpha_0,\theta_0}(\omega)$, just like the mean field ones, are holomorphic in the upper complex $\omega$-plane. This implies that the entries of the matrix $\Pi_{ij,\alpha_0,\theta_0}(\omega)$ do not have poles in the upper complex $\omega$-plane.

Let us now consider the following modulation of the system. We consider the protocol $(\alpha_s, \theta_s) = (\alpha_0 + \delta_\alpha(s), \theta_0 + \delta_\theta(s))$ and assume for $0 \leq s < \tilde{s}$ the system retains stability. For $(\alpha_{\tilde{s}}, \theta_{\tilde{s}}) = (\tilde{\alpha}, \tilde{\theta})$,

the system loses stability as $R$ poles $\omega_l$, $l = 1, \ldots, R$ cross into the upper complex $\omega$-plane (with $\mathbf{Im}\{\omega_l\} = 0$, $l = 1, \ldots, R$) for the macroscopic susceptibilities $\tilde{\Gamma}_{i,\tilde{\alpha},\tilde{\theta}}(\omega)$ (condition ii) is broken), while the mean field susceptibilies $\Gamma_{i,\tilde{\alpha},\theta}(\omega)$ are holomorphic in the upper complex $\omega$-plane (condition i) holds). This implies that the spectral gap of the operator $M^{0,+}_{\alpha,\theta,\langle x \rangle_0}$ is finite, so that there is no divergence of the integrated autocorrelation time of any observable.

We have that $P_{ij,\tilde{\alpha},\tilde{\theta}}$ does *not* have full rank for $\omega = \omega_l$, $l = 1, \ldots, R$. For such value(s) of $\omega$, *the macroscopic susceptibilities diverge.* Indeed, we remark that the invertibility conditions of the matrix $P_{ij,\alpha,\theta}(\omega)$ is intrinsic and does not depend on the applied external forcing $\mathbf{X}$, which enters, instead, only in the definition of the mean field susceptibility $\Gamma_{i,\alpha,\theta}(\omega)$. We interpret this as the fact that the divergence of the response is due to eminently endogenous, rather than exogenous, processes.

We also remark that $P_{ij,\alpha,\theta}(\omega) = \delta_{ij} - \Upsilon_{\{i,j\},\alpha,\theta}(\omega)$, where $\Upsilon_{\{i,j\},\alpha,\theta}(\omega)$ can be seen as mean field susceptibility for the expectation value of $x_i$ associated with an infinitesimal change of the value of the $j^{th}$ component of $\langle \mathbf{x} \rangle_0$, see equations (2.4) and (2.10). This supports the idea that $\langle \mathbf{x} \rangle$ is a appropriate order parameter for the system.

We assume, for simplicity, that only simple poles are present. We then decompose the matrix $\Pi_{ij,\tilde{\alpha},\tilde{\theta}}(\omega)$ in the upper complex $\omega$-plane as follows:

$$\Pi_{ij,\tilde{\alpha},\tilde{\theta}}(\omega) = \Pi^h_{ij,\tilde{\alpha},\tilde{\theta}}(\omega) + \sum_{l=1}^{R} \frac{\mathrm{Res}(\Pi_{ij,\tilde{\alpha},\tilde{\theta}}(\omega))_{\omega=\omega_l}}{\omega - \omega_l} \tag{4.1}$$

where we have separated the holomorphic component $\Pi^h_{ij,\tilde{\alpha},\tilde{\theta}}(\omega)$ from the singular contributions coming from the poles $\omega_l$, $l = 1, \ldots, R$; note that $\mathrm{Res}(f(\omega))_{\omega=\nu}$ indicates the residue of the function $f$ for $\omega = \nu$. Note that if $\omega_l$ is a pole on the real axis, $-\omega_l$ is also a pole. Additionally, $\mathrm{Res}(f(\omega))_{\omega=\omega_l} = -\mathrm{Res}(f(\omega))^*_{\omega=-\omega_l}$, so that if $\omega_l = 0$ the residue has vanishing real part.

Building on equation (4.1), the macroscopic susceptibility can then be written as:

$$\tilde{\Gamma}_{i,\tilde{\alpha},\tilde{\theta}}(\omega) = \Pi_{ij,\tilde{\alpha},\tilde{\theta}}(\omega)\Gamma_{i,\tilde{\alpha},\tilde{\theta}}(\omega) = \Pi^h_{ij}(\omega)\Gamma_{i,\tilde{\alpha},\tilde{\theta}}(\omega) + \sum_{l=1}^{R} \frac{\mathrm{Res}(\Pi_{ij,\tilde{\alpha},\tilde{\theta}}(\omega))_{\omega=\omega_l}}{\omega - \omega_l}\Gamma_{i,\tilde{\alpha},\tilde{\theta}}(\omega_l), \tag{4.2}$$

where the Kramers–Kronig relations given in equation (3.3) are then modified as follows, taking into account the extra poles along the real $\omega$-axis:

$$\mathbf{P}\int_{-\infty}^{\infty} d\nu \frac{\tilde{\Gamma}_{i,\tilde{\alpha},\tilde{\theta}}(\nu)}{\nu - \omega} = i\pi\,\tilde{\Gamma}_{i,\tilde{\alpha},\tilde{\theta}}(\omega) + i\pi\sum_{l=1}^{R}\frac{\mathrm{Res}(\Pi_{ij,\tilde{\alpha},\tilde{\theta}}(\omega))_{\omega=\omega_l}}{\omega_l - \omega}\Gamma_{i,\tilde{\alpha},\tilde{\theta}}(\omega_l). \tag{4.3}$$

By taking the limit $\omega \to \infty$ we can generalize the sum rule given in equation (3.7):

$$\int_0^{\infty} d\nu\,\mathbf{Re}\{\tilde{\Gamma}_{i,\tilde{\alpha},\tilde{\theta}}(\nu)\} = \frac{\pi}{2}\langle X_i(\mathbf{x})\rangle_0 - \frac{\pi}{2}\mathbf{Im}\left\{\sum_{l=1}^{R}\mathrm{Res}(\Pi_{ij,\tilde{\alpha},\tilde{\theta}}(\omega))_{\omega=\omega_l}\Gamma_{i,\tilde{\alpha},\tilde{\theta}}(\omega_l)\right\}. \tag{4.4}$$

Instead, by taking the limit $\omega \to 0$ we can generalize the sum rule given in equation (3.8) as follows:

$$\int_0^{\infty} d\nu\,\frac{\mathbf{Im}\{\tilde{\Gamma}_{i,\tilde{\alpha},\tilde{\theta}}(\nu)\}}{\nu} = \lim_{\omega\to 0}\left(\frac{\pi}{2}\mathbf{Re}\{\tilde{\Gamma}_{i,\tilde{\alpha},\tilde{\theta}}(\omega)\}\right) + \frac{\pi}{2}\mathbf{Re}\left\{\sum_{\omega_l\neq 0}\frac{\mathrm{Res}(\Pi_{ij,\tilde{\alpha},\tilde{\theta}}(\omega))_{\omega=\omega_l}}{\omega_l}\Gamma_{i,\tilde{\alpha},\tilde{\theta}}(\omega_l)\right\}, \tag{4.5}$$

where we note that the zero-frequency poles do not contribute to the second term on the right-hand side.

## (b) Two scenarios of phase transition

In the discussion above, we are assuming that for $(\alpha, \theta) = (\tilde{\alpha}, \tilde{\theta})$ we have that $\det(P_{ij,\alpha,\theta}(\omega))$ vanishes for $R$ real values of $\omega$. Since $P_{ij,\alpha,\theta}(\omega) = (P_{ij,\alpha,\theta}(-\omega^*))^*$, we have that $\det(P_{ij,\alpha,\theta}(\omega)) = (\det(P_{ij,\alpha,\theta}(-\omega^*)))^*$. Therefore, the solutions to the equation $\det(P_{ij,\alpha,\theta}(\omega)) = 0$ come in conjugate

pairs if they are complex. Generically, we can assume that as we tune the parameter $s$ to the critical value $\tilde{s}$ such that $(\alpha_{\tilde{s}}, \theta_{\tilde{s}}) = (\tilde{\alpha}, \tilde{\theta})$ either one real solution or the real part of one pair of solutions crosses to positive values. We then consider the following two scenarios for the poles $\omega_l, l = 1, \ldots, R$:

— $\omega_1 = 0$, $R = 1$; or
— $\omega_1 = -\omega_2 > 0$, $R = 2$.

Indeed, we wish to consider the two qualitatively different cases of either (i) a single pole with zero frequency; or (ii) a pair of poles with non-vanishing and opposite frequencies emerging at $(\alpha, \theta) = (\tilde{\alpha}, \tilde{\theta})$. Of course, more than two poles could simultaneously emerge $(\alpha, \theta) = (\tilde{\alpha}, \tilde{\theta})$, but we consider this as a non-generic case.

— If $\omega_l = 0$ is a pole, then we have a static phase transition, associated with a breakdown in the linear response describing the parametric modulation of the measure of the system, see §6a. While such a statement applies for rather general systems and perturbations, this situation can be better understood by considering the specific perturbation $\mathbf{X}(\mathbf{x}) = \langle \mathbf{x} \rangle_0 - \mathbf{x}$ with $T(t) = 1$, which amounts to studying, within linear approximation, how the measure of the system changes as the value of $\theta$ is changed to $\theta + \epsilon$. This phase transition corresponds to a insulator-metal phase transition in condensed matter, because the electric susceptibility $\chi_{ij}^{(1)}(\omega)$ of a conductor diverges as $i\sigma_{ij}/\omega$ for small frequencies, where $\sigma$ is a real tensor and describes the static electric conductivity, which is vanishing for an insulator [30].
— If, instead, we have a pair of poles located at $\pm \omega_l \neq 0$, we have a dynamic phase transition activated by a forcing with non-vanishing spectral power at the frequency $\pm \omega_l$. In this case, a limit cycle emerges corresponding to self-sustained oscillation, which is made possible by the feedback encoded in the nonlinearity of the McKean–Vlasov equation, see e.g. [89] and §6b.

In §6, we will present examples of phase transitions occurring according to the two scenarios above.

## 5. Equilibrium phase transitions: gradient systems

When the local force can be written as a gradient of a potential $\mathbf{F}_\alpha(\mathbf{y}) = -\nabla V_\alpha(\mathbf{y})$ and the diffusion matrix is the identity matrix $s_{ij} = \delta_{ij}$, equations (2.1) describe an equilibrium system. In particular, the $N$ particles system has a unique ergodic invariant measure when the potential satisfies suitable confining properties [16,63] (see later discussion). Equivalently, the generator of the finite particle stochastic process has purely discrete spectrum, a non-zero spectral gap and the system converges exponentially fast to the unique equilibrium state, both in the $L^2$ space weighted by the invariant measure and in relative entropy.

In the limit $N \to \infty$, the system is described by the McKean–Vlasov equation (2.2) whose stationary measures are solutions of the Kirkwood–Monroe equation [105]:

$$\rho_{\alpha,\theta}^{(0)} = \frac{1}{Z} e^{-(2/\sigma^2)\left(V(\mathbf{x}) + U \star \rho_{\alpha,\theta}^{(0)}(\mathbf{x})\right)}, \quad Z = \int e^{-(2/\sigma^2)\left(V(\mathbf{x}) + U \star \rho_{\alpha,\theta}^{(0)}(\mathbf{x})\right)} \, d\mathbf{x}. \tag{5.1}$$

When the confining and interaction potentials are strongly convex and convex, respectively, then it is well known that equation (5.1) has only one solution, corresponding to the unique steady state of the McKean–Vlasov dynamics [106]. In addition, the dynamics converges exponentially fast, in relative entropy, to the stationary state and the rate of convergence to equilibrium can be quantified [106]. However, when the confining potential is not convex, e.g. is bistable, then more than one stationary states can exist, at sufficiently low noise strength (equivalently, for sufficiently strong interactions). A well-known example where the non-uniqueness of the invariant measure

is that of the Desai–Zwanzig model [1,86,103], where the interaction potential is quadratic (see §6a for more details). In this framework, the loss of uniqueness of the invariant measure can be interpreted as a continuous-phase transition, similar to some extent to the phase transition for the Ising model. For a quadratic interaction potential, the equilibrium stationary measure (5.1) can be written as

$$\rho_{\alpha,\theta}^{(0)} = \frac{1}{Z}\, e^{-(2/\sigma^2)\hat{V}}, \quad Z = \int e^{-(2/\sigma^2)\hat{V}}\, d\mathbf{x}, \tag{5.2}$$

where we have introduced the modified potential $\hat{V}(\mathbf{x}) = V(\mathbf{x}) - \theta((|\mathbf{x}|^2/2) - \langle \mathbf{x}\rangle_0 \cdot \mathbf{x})$, with the term proportional to $\theta$ arising from the interactions between the subsystems. The linear Fokker–Planck operator associated to the stationary Mc-Kean Vlasov equation (2.2) describing the equilibrium dynamics relative to (5.2) reads

$$M_{\alpha,\theta,\langle\mathbf{x}\rangle_0}^0(\cdot) = \nabla \cdot \left( \nabla \hat{V}(\mathbf{x}) \cdot \right) + \frac{\sigma^2}{2}\Delta. \tag{5.3}$$

It is well known [63, Sect. 4.5] that, if the modified potential $\hat{V}$ satisfies the property

$$\lim_{|\mathbf{x}|\to+\infty} \left( \frac{|\nabla\hat{V}|^2}{2} - \Delta\hat{V} \right) = +\infty \tag{5.4}$$

then the operator $M_{\alpha,\theta,\langle\mathbf{x}\rangle_0}^0$ in (5.3) has a spectral gap in $L^2(\rho_{\alpha,\theta}^0)$, the space of square integrable functions weighted with by the invariant density $\rho_{\alpha,\theta}^0$. In particular, condition (5.4) prevents the system from undergoing a phase transition via scenario (i). When detailed balance holds, the mean field susceptibility $G_{i,\alpha,\theta}(\tau)$ relative to a uniform spatial forcing $\mathbf{X} = \text{const.}$ can be written as the time derivative of suitable correlation functions. In fact, from equation (2.11), the mean field susceptibility can be written as

$$G_{i,\alpha,\theta}(\tau) = -\Theta(\tau) \int d^M \mathbf{y}\, y_i \exp\left( M_{\alpha,\theta,\langle\mathbf{x}\rangle_0}^0 \tau \right) \nabla \cdot \left( \rho_{\alpha,\theta}^{(0)}(\mathbf{y})\mathbf{X}(\mathbf{y}) \right). \tag{5.5}$$

Without loss of generality, let us consider an uniform forcing $\mathbf{X} = \hat{\mathbf{v}}_k$, with $\hat{\mathbf{v}}_k$ being the unit vector in the $k$th direction. The mean field susceptibility thus becomes

$$G_{i,\alpha,\theta}(\tau) = -\Theta(\tau) \int d^M \mathbf{y}\, y_i \exp\left( M_{\alpha,\theta,\langle\mathbf{x}\rangle_0}^0 \tau \right) \partial_{y_k} \rho_{\alpha,\theta}^{(0)} = \frac{\Upsilon_{\{i,k\},\alpha,\theta}(\tau)}{\theta}. \tag{5.6}$$

Since the system is at equilibrium and the stationary probability density can be written as in (5.1), $\partial_{y_k}\rho_{\alpha,\theta}^{(0)} = -(2/\sigma^2)\rho_{\alpha,\theta}^{(0)}\partial_{y_k}\hat{V}$, physically representing the fact that the probability current associated with the invariant measure vanishes at equilibrium. Furthermore, using (5.3) it is easy to verify the following identity $M_{\alpha,\theta,\langle\mathbf{x}\rangle_0}^0(y_k\rho_{\alpha,\theta}^{(0)}) = -\rho_{\alpha,\theta}^{(0)}\partial_{y_k}\hat{V}$. The mean field susceptibility can then be written as

$$G_{i,\alpha,\theta}(\tau) = -\frac{2}{\sigma^2}\Theta(\tau) \int d^M \mathbf{y}\, y_i \exp\left( M_{\alpha,\theta,\langle\mathbf{x}\rangle_0}^0 \tau \right) M_{\alpha,\theta,\langle\mathbf{x}\rangle_0}^0 y_k \rho_{\alpha,\theta}^{(0)} \tag{5.7}$$

$$= -\frac{2}{\sigma^2}\Theta(\tau)\frac{d}{d\tau} \int d^M \mathbf{y}\, y_i \exp\left( M_{\alpha,\theta,\langle\mathbf{x}\rangle_0}^0 \tau \right) y_k \rho_{\alpha,\theta}^{(0)} \tag{5.8}$$

$$= -\frac{2}{\sigma^2}\Theta(\tau)\frac{d}{d\tau}\langle x_i(\tau)x_k(0)\rangle_0 \tag{5.9}$$

$$= -\frac{2}{\sigma^2}\Theta(\tau)\frac{d}{d\tau}\langle z_i(\tau)z_k(0)\rangle_0, \tag{5.10}$$

where in the last equation we have introduced the fluctuation variables $z_i = x_i - \langle x_i\rangle_0$. Equation (5.10) shows that the mean field susceptibility is closely related to equilibrium

correlation functions. It is then possible to associate to each correlation function the correlation time

$$\tau_{ij} = \frac{\int_0^{+\infty} dt \langle z_i(\tau) z_j(0) \rangle_0}{\langle z_i(0) z_j(0) \rangle_0}. \tag{5.11}$$

Note that this time scale differs from the one introduced in equation (3.8), which in this case can be written as

$$\tau_{G_i} = \frac{\int_0^{\infty} dt\, G_{i,\alpha,\theta}(t)}{G_{i,\alpha,\theta}(0^+)} = -\frac{\langle z_i(0) z_j(0) \rangle_0}{\lim_{t \to 0+} d/dt \langle z_i(t) z_j(0) \rangle_0}.$$

By comparing the expressions of $\tau_{G_i}$ and $\tau_{ij}$ and by considering equation (2.15), one understands that $\tau_{G_i}$ and $\tau_{ij}$ correspond to two differently weighted averages of the timescales associated with each subdominant mode of the operator $M^0_{\alpha,\theta,\langle \mathbf{x} \rangle_0}$.

Usually, the singular behaviour of correlation properties has been used as an indicator of critical transitions [74]. However, let us remark again that, being related to the spectrum of the operator $M^0_{\alpha,\theta,\langle x_0 \rangle}$, in our case neither $\tau_{G_i}$ nor $\tau_{ij}$ show any critical behaviour at transitions occurring according to the scenario (ii), while they both diverge in the case of critical transitions corresponding to the scenario (i).

# 6. Examples

In what follows we re-examine the linear response of two relevant models that have been extensively investigated in the literature. Using the framework developed above, we investigate the phase transitions occurring in the Desai–Zwanzig model [86] and the Bonilla–Casado–Morilla model [89], which are taken as paradigmatic examples of equilibrium and non-equilibrium systems, respectively. We also provide the result of numerical simulations for both models.

## (a) Equilibrium phase transition: the Desai–Zwanzig model

The Desai–Zwanzig model [86] has a paradigmatic value as it features an equilibrium thermodynamic phase transition (pitchfork bifurcation) arising from the interaction between systems [107] and has been used also as a model for systemic risk [3]. Each of the systems can be interpreted as a particle, moving in one dimension ($M = 1$) in a double well potential $V_\alpha(x) = -(\alpha/2)x^2 + x^4/4$, interacting with the other particles via a quadratic interaction $U(x)$. The $N-$ particle system is described by

$$dx^k = F_\alpha(x^k)dt - \frac{\theta}{N} \sum_{l=1}^{N} \partial_{x^k} U(x^k - x^l)dt + \sigma dW^k, \tag{6.1}$$

where $k = 1, \ldots, N$. The local force is $F_\alpha = -V'_\alpha$, the interaction potential is $U(x) = x^2/2$ and the volatility matrix is the identity matrix $s_{ij} = \delta_{ij}$. Furthermore, $V_\alpha$ is double-well-shaped when $\alpha > 0$, otherwise it has a unique global minimum. In the thermodynamic limit $N \to \infty$, the one particle density satisfies the McKean–Vlasov equation (2.2) and it has been proven [1,103] that the infinite particle system undergoes a continuous phase transition, with $\langle x \rangle$ being a suitable order parameter. The Desai–Zwanzig model can be seen as a stochastic model of key importance for elucidating order–disorder phase transitions [107].

We have studied the Desai–Zwanzig model also through numerical integration of equation (6.1) by adopting an Euler–Maruyama scheme [108]. We have tested the convergence of our results in the thermodynamic limit $N \to \infty$ by looking at increasing values of the number $N$ of particles. We present in figures 1a–c the results obtained with $N = 5000$ for $0.2 \le \theta \le 1.0$ and $0.4 \le \sigma \le 1.0$. The relevant expectation values and correlations have been evaluated considering averages performed over $2.5 \times 10^3$ time units. Figure 2a,b portrays two sections performed approximately in the middle of the domain of the heat maps provided in figure 1a–c, with the goal of clarifying the obtained results. The order parameter clearly indicates a continuous-phase transition. The re-scaled variance of the fluctuations, being related to the operator $M^0_{\alpha,\theta,\langle x \rangle_0}$, is

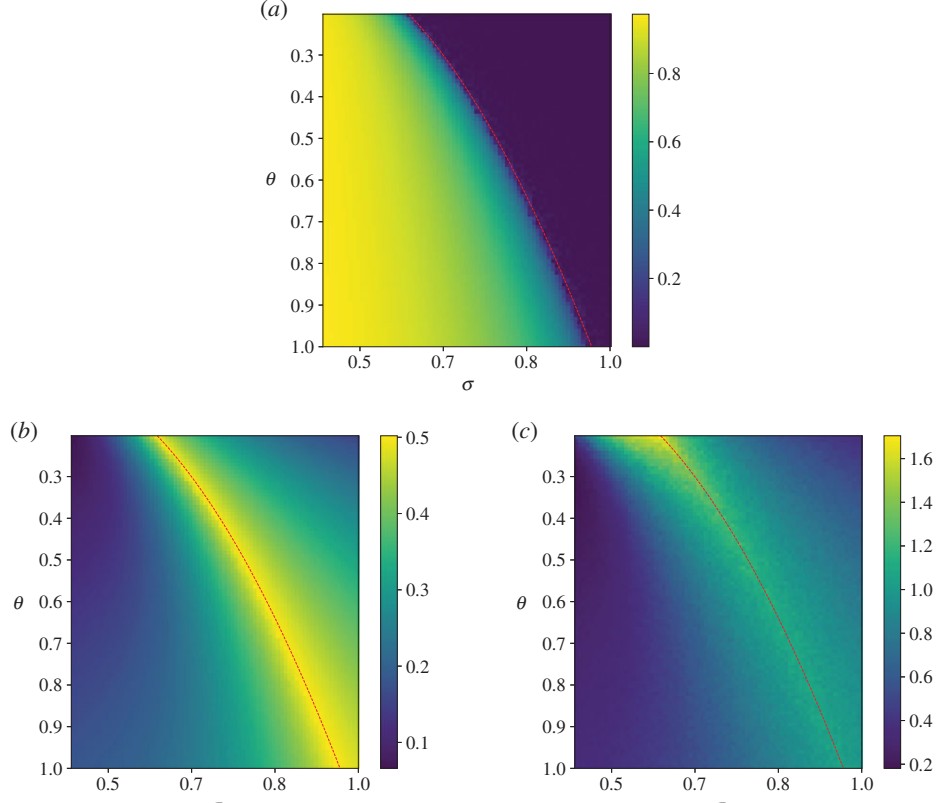

**Figure 1.** Results of numerical simulations of equation (6.1) with $\alpha = 1$. Heat maps of the order parameter $\langle x \rangle_0$ (panel (*a*)); of the re-scaled variance $(\theta/\sigma^2)\langle z^2 \rangle_0$ (panel (*b*)); and of the rescaled correlation time $\hat{\tau} = \theta \times \tau$ (panel (*c*)). The dotted red line shows the transition line, see [1]. See text for details. (Online version in colour.)

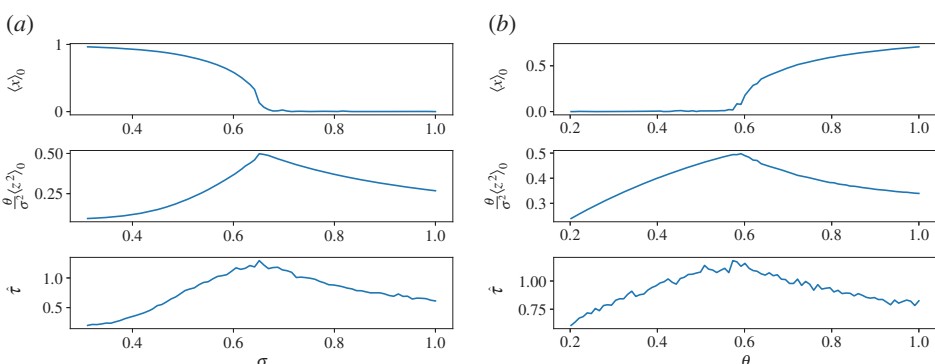

**Figure 2.** A horizontal (left) and a vertical (right) section of the heat maps shown in figure 1*a–c*. (*a*) From top to bottom: order parameter, rescaled variance and rescaled integrated autocorrelated time as a function of the strength of the noise. Here $\theta \approx 0.4$. (*b*) From top to bottom: order parameter, rescaled variance and rescaled integrated autocorrelated time as a function of the strength of the coupling. Here $\sigma \approx 0.78$. (Online version in colour.)

finite (and equal to 1/2) at the transition point, in agreement with equation (6.8). The re-scaled correlation time $\hat{\tau} = \theta \times \tau$, where $\tau$ is defined in (5.11), is also non-singular, as discussed below. The response of the order parameter to a perturbation $F_\alpha \to F_\alpha + \varepsilon X(x)T(t)$ is given by

equation (2.8). Given the simplicity of this model, it is possible to explicitly evaluate all the relevant quantities that characterize a phase transition relative to scenario (ii). Indeed, if we consider a perturbation such that $X(x) = 1$, one has $Y_{\alpha,\theta}(\tau) = \theta G_{\alpha,\theta}(\tau)$ and equation (2.16) can be written as

$$P(\omega)\langle x \rangle_1(\omega) = \Gamma_{\alpha,\theta}(\omega)T(\omega) \tag{6.2}$$

where the $1 \times 1$ matrix is $P(\omega) = 1 - \theta \Gamma_{\alpha,\theta}(\omega)$. The macroscopic susceptibility is then obtained as

$$\tilde{\Gamma}_{\alpha,\theta}(\omega) = P^{-1}(\omega)\Gamma_{\alpha,\theta}(\omega) = \frac{\Gamma_{\alpha,\theta}(\omega)}{1 - \theta \Gamma_{\alpha,\theta}(\omega)}. \tag{6.3}$$

Furthermore, this is a gradient system satisfying all the assumptions that have been made in §5, so that the mean field susceptibility can be written as (see also [103])

$$G_{\alpha,\theta}(\tau) = -\Theta(\tau)\frac{2}{\sigma^2}\frac{d}{d\tau}\langle z(\tau)z(0)\rangle_0, \tag{6.4}$$

where $z(t) = x - \langle x \rangle_0$. Taking the Fourier transform results in

$$\Gamma_{\alpha,\theta}(\omega) = \frac{2}{\sigma^2}\left[\langle z^2 \rangle_0 - i\omega\gamma(\omega)\right]. \tag{6.5}$$

where $\gamma(\omega) = \int_0^\infty dt e^{-i\omega t}\langle z(t)z(0)\rangle_0$ is the Fourier transform of the correlation function. As previously mentioned, $\Gamma_{\alpha,\theta}(\omega)$ can be written in terms of the spectrum of the operator $M_{\alpha,\theta,\langle x \rangle_0}^{0,+}$ which in this specific example reads (see equation (5.3))

$$M_{\alpha,\theta,\langle x \rangle_0}^{0,+} = -\hat{V}'(x)\partial_x + \frac{\sigma^2}{2}\partial_{xx}, \tag{6.6}$$

where the modified potential is $\hat{V} = V_\alpha - \theta(x^2/2 - \langle x \rangle_0 x)$. It can be proven [1] that the above operator is self-adjoint and has a pure point spectrum $\{\lambda_\mu\}$ with $0 = \lambda_0 > \lambda_1 > \lambda_2 > \ldots$, with the vanishing eigenvalue corresponding to the stationary distribution $\rho_{\alpha,\theta}^{(0)}$. In fact, it is easy to show that condition (5.4) holds. The operator $\tilde{M}_{\alpha,\theta,\langle x \rangle_0}^{0,+}$ is instead

$$\tilde{M}_{\alpha,\theta,\langle x \rangle_0}^{0,+}(\rho_{\alpha,\theta}^{(1)}) = M_{\alpha,\theta,\langle x \rangle_0}^{0,+}(\rho_{\alpha,\theta}^{(1)}) - \theta\langle x \rangle_1(t)\partial_x\rho_{\alpha,\theta}^{(0)}. \tag{6.7}$$

Dawson [1] proved that, away from the transition point—in particular, above it, where $\langle x \rangle_0 = 0$—the nonlinear operator $\tilde{M}_{\alpha,\theta,\langle x \rangle_0}^{0,+}$ has similar spectral properties to $M_{\alpha,\theta,\langle x \rangle_0}^{0,+}$. At the transition, though, $\tilde{M}_{\alpha,\theta,\langle x \rangle_0}^{0,+}$ shows a vanishing spectral gap, with the operator developing a null eigenvalue. This situation corresponds to the breakdown of the aforementioned condition (ii) in which the mean field susceptibility $\Gamma_{\alpha,\theta}(\omega)$—and thus $\gamma(\omega)$—is holomorphic in the upper complex $\omega$-plane, while the macroscopic $\tilde{\Gamma}_{\alpha,\theta}$ develops a pole, arising from the non-invertibility of $P(\omega)$. Let us observe again that this implies that at the transition there is no divergence of the integrated autocorrelation time $\tau$, because the spectral gap of the operator $M_{\alpha,\theta,\langle x \rangle_0}^{0,+}$ does not shrink to zero. This is clearly shown in the two-dimensional map shown in figure 1c and in the two sections shown in figure 2a,b. We can fully characterize the singular behaviour of the macroscopic susceptibility $\tilde{\Gamma}_{\alpha,\theta}$ at the transition. As a matter of fact, the transition point is characterized [103] by the condition

$$1 - \frac{2\theta}{\sigma^2}\langle z^2 \rangle_0 = 0 \tag{6.8}$$

so that the macroscopic susceptibility becomes

$$\tilde{\Gamma}_{\alpha,\theta}(\omega) = \frac{(2/\sigma^2)\left[\langle z^2 \rangle_0 - i\omega\gamma(\omega)\right]}{i\theta\omega\gamma(\omega)} = -\frac{2}{\theta\sigma^2} + \frac{2\langle z^2 \rangle_0}{i\theta\omega\gamma(\omega)} \tag{6.9}$$

As previously discussed in relation to equation (4.2), the above expression shows that at the transition point $\tilde{\Gamma}_{\alpha,\theta}$ develops a simple pole in $\omega = 0$, with residue

$$\text{Res}(\tilde{\Gamma}_{\alpha,\theta})_{\omega=0} = -i\frac{2\langle z^2 \rangle_0}{\theta\gamma(0)} \tag{6.10}$$

## (b) Non-equilibrium phase transition: the Bonilla–Casado–Morilla model

In this section, we will study the Bonilla–Casado–Morrillo model [89] and elucidate the properties of a non-equilibrium self-synchronization phase transition, by looking at the divergence of the macroscopic susceptibility $\tilde{\Gamma}_{\alpha,\theta}$. We anticipate that the susceptibility develops a pair of symmetric poles $\omega_1 = -\omega_2 > 0$ at the transition point, thus following the scenario (ii) discussed above. The model consists of $N$ two-dimensional nonlinear oscillators $\mathbf{x}^k = (x_1^k, x_2^k)$, interacting via a quadratic interaction potential $U(\mathbf{x}) = |\mathbf{x}|^2/2$ and subjected to thermal noise

$$\mathrm{d}x_i^k = F_{i,\alpha}(\mathbf{x}^k)\mathrm{d}t - \frac{\theta}{N}\sum_{l=1}^{N}\partial_{x_i^k}U(\mathbf{x}^k - \mathbf{x}^l)\mathrm{d}t + \sigma\,\mathrm{d}W_i^k, \quad k = 1,\ldots,N. \tag{6.11}$$

The local force is not conservative, giving rise to a non-equilibrium process, and reads $\mathbf{F}_\alpha(\mathbf{x}) = (\alpha - |\mathbf{x}|^2)\mathbf{x} + \mathbf{x}^+$ where $\mathbf{x}^+ = (-x_2, x_1)$. This term corresponds to a rotation, which is divergence-free with respect to the (Gibbsian) invariant measure and, therefore, does not change the stationary state, but it makes it a non-equilibrium one [109–111]. The systematic study of linear response theory for such non-equilibrium systems is an interesting problem that we leave for future study. In the thermodynamic limit, the system is described by a McKean–Vlasov equation

$$\partial_t\rho(\mathbf{x},t) = -\nabla \cdot \left[\left(\hat{\mathbf{F}} + \theta\langle\mathbf{x}\rangle\right)\rho\right] + \frac{\sigma^2}{2}\Delta\rho, \tag{6.12}$$

where $\hat{\mathbf{F}} = \mathbf{F}_\alpha - \theta\mathbf{x}$, the last term representing the mean field contribution of the coupling to the local force. The authors in [89] prove that the infinite particle system undergoes a phase transition, with a stationary measure $\rho_0(\mathbf{x})$ losing stability to a time-dependent probability measure $\bar{\rho} = \bar{\rho}(\mathbf{x},t)$. Physically, this phenomenon can be interpreted as a process of synchronization. In fact, $\rho^{(0)}(\mathbf{x})$ represents a disordered state, with the oscillators moving out of phase, while $\bar{\rho}$ describes a state of collective organization with the oscillators moving in an organized rhythmic manner. The transition can be investigated via the order parameter $\langle\mathbf{x}\rangle$ which vanishes in the asynchronous state, $\langle\mathbf{x}\rangle_0 = 0$, and is different from zero and time dependent in the synchronous state. In particular, the stationary measure $\rho_0(\mathbf{x})$ can be written as

$$\rho^{(0)}(\mathbf{x}) = \frac{1}{Z}\,\mathrm{e}^{-\phi(\mathbf{x})}, \quad \phi(\mathbf{x}) = \left(\theta - \alpha + \frac{1}{2}|\mathbf{x}|^2\right)\frac{|\mathbf{x}|^2}{\sigma^2} \tag{6.13}$$

and satisfies the stationary McKean–Vlasov equation

$$M_{\alpha,\theta,\langle x\rangle_0}^0\left(\rho^{(0)}\right) = 0 \tag{6.14}$$

with $M_{\alpha,\theta,\langle x\rangle_0}^0(g) = -\nabla \cdot [\hat{F}g] + (\sigma^2/2)\Delta g$, being the Fokker–Planck operator describing the stationary state $\rho^{(0)}(\mathbf{x})$. Note that $\langle x_0\rangle = 0$. We can perform a linear response theory around this stationary state $\rho_0$ by replacing $\mathbf{F}_\alpha \to \mathbf{F}_\alpha + \varepsilon\mathbf{X}(\mathbf{x})T(t)$ and studying the perturbation $\rho^{(1)}$ of the measure defined via $\rho(\mathbf{x},t) = \rho^{(0)}(\mathbf{x}) + \varepsilon\rho^{(1)}(\mathbf{x},t)$. As previously outlined, $\rho^{(1)}(\mathbf{x},t)$ satisfies equation (2.4) from which the whole linear response theory follows. However, to conform to the notation in [89] we will here define $\rho^{(1)}(\mathbf{x},t) = (\rho^{(0)})^{1/2}q(\mathbf{x},t)$ and write the corresponding equation for $q(\mathbf{x},t)$. After some algebra, it is possible to write that

$$\partial_t q(\mathbf{x},t) = \mathcal{M}_{\alpha,\theta,0}(q) - T(t)(\rho^{(0)})^{-1/2}\nabla \cdot \left(\mathbf{X}(\mathbf{x})\rho^{(0)}\right) + \theta(\rho^{(0)})^{1/2}\langle(\rho^{(0)})^{1/2}\mathbf{y}, q(\mathbf{y},t)\rangle \cdot \nabla\phi(\mathbf{x})$$

$$= \tilde{\mathcal{M}}_{\alpha,\theta,0}(q) - T(t)(\rho^{(0)})^{-1/2}\nabla \cdot \left(\mathbf{X}(\mathbf{x})\rho^{(0)}\right) \tag{6.15}$$

where we have defined

$$\mathcal{M}_{\alpha,\theta,0}(q) = \frac{\sigma^2}{4}\left[\Delta\phi - \frac{1}{2}|\nabla\phi|^2\right]q + \left[-\mathbf{x}^+ \cdot \nabla + \Delta\right]q \tag{6.16}$$

We mention that operator has the structure of a Schrödinger operator in a magnetic field ([63], Sec. 4.9). Furthermore, $\tilde{\mathcal{M}}_{\alpha,\theta,0}(q) = \mathcal{M}_{\alpha,\theta,0}(q) + \theta(\rho^{(0)})^{1/2}\langle(\rho^{(0)})^{1/2}\mathbf{y}, q(\mathbf{y},t)\rangle \cdot \nabla\phi(\mathbf{x})$ with $\langle f, g\rangle = \int \mathrm{d}\mathbf{y}f(\mathbf{y})g(\mathbf{y})$ being the usual scalar product. In particular, let us observe that

$\langle(\rho^{(0)})^{1/2}\mathbf{y}, q(\mathbf{y}, t)\rangle = \int(\rho^{(0)})^{1/2}\mathbf{y}q(\mathbf{y}, t)d\mathbf{y} = \int \mathbf{y}\rho^{(1)}(\mathbf{y}, t)d\mathbf{y} = \langle\mathbf{y}\rangle_1$. A formal solution of the above equation is

$$q(\mathbf{x}, t) = \int_{-\infty}^{t} ds \exp\left[\mathcal{M}_{\alpha,\theta,0}(t-s)\right]\left[-T(s)(\rho_{(0)})^{-1/2}\nabla \cdot \left(\mathbf{X}(\mathbf{x})\rho^{(0)}\right)\right.$$
$$\left.+\theta(\rho^{(0)})^{1/2}\langle(\rho^{(0)})^{1/2}\mathbf{y}, q(\mathbf{y}, s)\rangle \cdot \nabla\phi(\mathbf{x})\right] \tag{6.17}$$

which is the analogous of equation (2.5). Using the above expression, we can evaluate the response of the observable $x_i$ as

$$\langle x_i\rangle_1 = \left\langle(\rho^{(0)})^{1/2}x_i, q(\mathbf{x}, t)\right\rangle = \int d\mathbf{x}\int_{-\infty}^{t} ds \quad x_i \exp\left[\mathcal{M}_{\alpha,\theta,0}(t-s)\right]$$
$$\times \left[-T(s)(\rho^{(0)})^{-1/2}\nabla \cdot \left(\mathbf{X}(\mathbf{x})\rho^{(0)}\right) + \theta(\rho^{(0)})^{1/2}\langle(\rho^{(0)})^{1/2}\mathbf{y}, q(\mathbf{y}, s)\rangle \cdot \nabla\phi(\mathbf{x})\right]. \tag{6.18}$$

Comparing equation (6.18) and (2.6), it is clear that the operators $\mathcal{M}_{\alpha,\theta,0}, \tilde{\mathcal{M}}_{\alpha,\theta,0}$ are analogous to the operators $M^0_{\alpha,\theta,\langle\mathbf{x}\rangle_0}, \tilde{M}^0_{\alpha,\theta,\langle\mathbf{x}\rangle_0}$ defined in §2. In particular, their spectrum is related to the Fourier transform of the mean field susceptibility $\Gamma_{\alpha,\theta}$ and macroscopic susceptibility $\tilde{\Gamma}_{\alpha,\theta}$ (respectively) through equations similar to (2.17) and (2.27). The authors in [89] study the spectrum of both these operators in order to perform a stability analysis of the stationary distribution $\rho_0(\mathbf{x})$. In particular, they observe that the operator $\mathcal{M}_{\alpha,\theta,0}$ can be written as $\mathcal{M}_{\alpha,\theta,0} = \mathcal{M}_H + \mathcal{M}_A$ where

$$\mathcal{M}_H(q) = \frac{\sigma^2}{4}\left[\Delta\phi - \frac{1}{2}|\nabla\phi|^2\right]q + \frac{\sigma^2}{2}\Delta q \tag{6.19}$$

and

$$\mathcal{M}_A(q) = -\mathbf{x}^+ \cdot \nabla q \tag{6.20}$$

with vanishing commutator $[\mathcal{M}_H, \mathcal{M}_A] = 0$. The operator $\mathcal{M}_H$ is related to the conservative part of the local force. As a matter of fact, it is a self-adjoint (Hermitian) operator with real eigenvalues. $\mathcal{M}_A$ is instead anti-Hermitian, with purely imaginary eigenvalues (describing oscillations) given by the non-conservative part of $\mathbf{F}$. Furthermore, $\mathcal{M}_H$ has only one zero eigenvalue corresponding to the ground state $(\rho^{(0)})^{1/2}$ while all the remaining eigenvalues are negative, meaning that scenario (i) in §4b, according to which the spectral gap of the mean field operator $M^0_{\alpha,\theta,\langle\mathbf{x}\rangle_0}$ vanishes, cannot happen in this setting. In particular, correlation properties will never diverge. Phase transition can, instead, take place according to the scenario (ii) above. Indeed, the authors in [89] show that the spectral gap of the operator $\tilde{\mathcal{M}}_{\alpha,\theta,0}$ vanishes at surface in the $(\alpha, \sigma, \theta)$ parametric space defined by the following equation:

$$A = \frac{\delta^2}{2}\left[1 - \frac{1}{\delta}\exp\left(-\frac{A^2}{\delta^2}\right)\left[\int_{-(A/\delta)}^{\infty}e^{-r^2}dr\right]^{-1}\right], \tag{6.21}$$

where $A = \alpha/\theta - 1$ and $\delta = \sqrt{2\sigma^2}/\theta$. In particular, they are able to prove that the eigenvalues associated with eigenfunctions of $\tilde{\mathcal{M}}_{\alpha,\theta,0}$ which are orthogonal to the subspace of $L^2(\mathbb{R}^2)$ spanned by $(\rho^{(0)})^{1/2}$ and $\mathbf{n} \cdot \mathbf{x}(\rho^{(0)})^{1/2}$, $\mathbf{n} \in \mathbb{R}^2$ being any unit vector, are always negative. Nevertheless, $\tilde{\mathcal{M}}_{\alpha,\theta,0}$ can become unstable from eigenfunctions which are not orthogonal to $\mathbf{n} \cdot \mathbf{x}(\rho^{(0)})^{1/2}$. It is possible to identify the eigenfunctions that at the transition yield eigenvalues with vanishing real part. In particular, at the transition line (6.21), the eigenfunction $\Omega(\mathbf{x}) = (0, 1) \cdot \mathbf{x}(\rho^{(0)})^{1/2} + i(1, 0) \cdot \mathbf{x}(\rho^{(0)})^{1/2}$ gives an eigenvalue $\tilde{\lambda}_j = i$, with $\Omega(\mathbf{x})^*$ corresponding to the complex conjugate eigenvalue $\tilde{\lambda}_j^* = -i$. The macroscopic susceptibility (2.27) consequently develops a pair of symmetric poles in $\omega = \pm 1$, corresponding to a dynamic phase transition, giving rise to a Hopf-like bifurcation yielding the time-dependent state $\bar{\rho}(\mathbf{x}, t)$ that defines the synchronized state. As a result, near the transition, the order parameter $\langle x\rangle$, where the expectation value is computed using the measure $\bar{\rho}(\mathbf{x}, t)$, oscillates at frequency $\omega = 1$ with amplitude $A_1(\alpha, \sigma, \theta)$. Instead, since it is a quadratic quantity, the rescaled variance $\theta/\sigma^2\langle z^2\rangle$, where $z = x - \langle x\rangle$, oscillates at frequency $\omega = 2$ with amplitude $A_2(\alpha, \sigma, \theta)$ around the value $B_2(\alpha, \sigma, \theta)$.

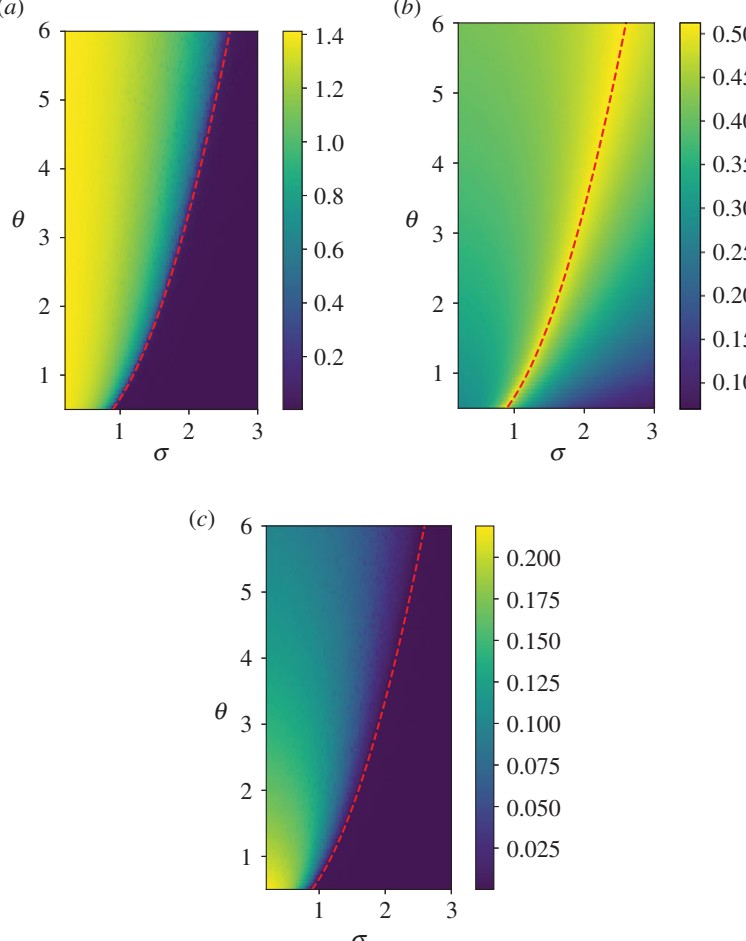

**Figure 3.** Results of numerical simulations of equation (6.11) with $\alpha = 2$. Heat maps of the amplitude $A_1$ of the oscillations of the variable $x$ (panel ($a$)), of the amplitude $A_2$ of the oscillations of the re-scaled variance $\theta/\sigma^2 \langle z^2 \rangle$ (panel ($b$)), and of the time mean value of $\theta/\sigma^2 \langle z^2 \rangle$ (panel ($c$)). The red dotted line represents the transition line given by equation (6.21); see [89]. See text for details. (Online version in colour.)

We have investigated this non-equilibrium transition through numerical integration of equation (6.11) via an Euler–Maruyama scheme [108]. The convergence of our results to the thermodynamic limit has been tested by looking at increasing values of the number of agents. We display here the results by taking $N = 5000$ and choosing $\alpha = 2$. Figure 3 shows the value of $A_1(\alpha = 2, \sigma, \theta)$ (panel $a$), $A_2(\alpha = 2, \sigma, \theta)$ (panel $b$) and $B_2(\alpha = 2, \sigma, \theta)$ (panel $c$) in the parametric region $0.2 \leq \sigma \leq 3, 0.5 \leq \theta \leq 6$ of the two dimensional parameter space $(\sigma, \theta)$. For the sake of clarity, we also provide in figure 4 a snapshot of a horizontal and vertical section of the heat plots.

These numerical experiments confirm that the system indeed undergoes a continuous phase transition, with a collective synchronization stemming from a disordered state as the system passes through the transition line given by equation (6.21) for $\alpha = 2$. Let us remark again that the fluctuations, being related to the spectrum of $M^{0,+}_{\alpha,\theta,\langle x \rangle_0}$, are always finite, see figure 4.

## 7. Conclusion

The understanding of how a network of exchangeable interacting systems responds to perturbations is a problem of great relevance in mathematics, natural and social sciences, and

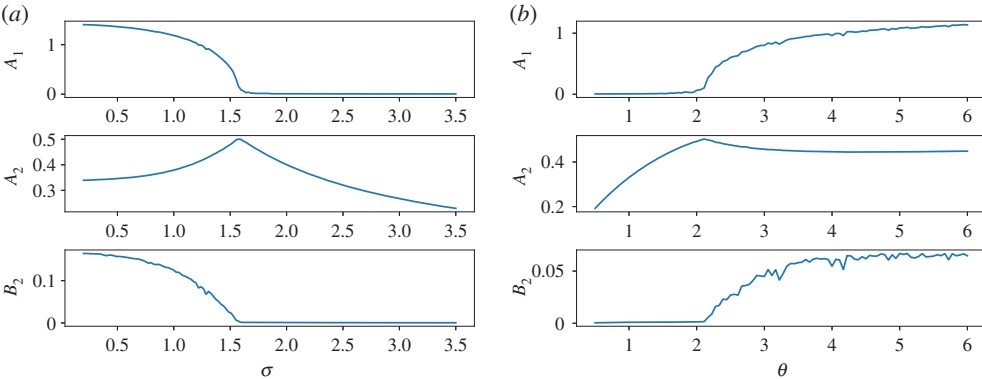

**Figure 4.** Horizontal (left) and vertical (right) sections of the heat plots 3*a*–*c*. (*a*) From top to bottom: $A_1(\alpha = 2, \sigma, \theta = 2)$, $A_2(\alpha = 2, \sigma, \theta = 2)$, and $B_2(\alpha = 2, \sigma, \theta = 2)$. (*b*) From top to bottom: $A_1(\alpha = 2, \sigma = 1.6, \theta)$, $A_2(\alpha = 2, \sigma = 1.6, \theta)$, and $B_2(\alpha = 2, \sigma = 1.6, \theta)$. (Online version in colour.)

technology. One is in general interested in both the smooth regime of response, where small perturbations result into small changes in the properties of the system, and in the non-smooth regime, which anticipates the occurrence of critical, possibly undesired, changes. Often, critical phenomena, which can be triggered by exogeneous or endogenous processes, are accompanied by the existence of a large-scale restructuring of the system, whereby spatial (i.e. across systems) and temporal correlations are greatly enhanced. The emergence of a specific spatial structure is especially clear when considering order–disorder transitions. Spatial–temporal coordination becomes evident when studying the multi-faceted phenomenon of synchronization. Finally, slow decay of temporal correlations—the so-called slowing down—indicates that nearby critical transitions the negative feedback of the system become ineffective.

This paper is the first step in a research programme that aims at developing practical tools for better understanding and predicting—in a data-driven framework—critical transitions in complex systems. We have here developed a fairly general theory of linear response for such a network in the thermodynamic limit of an infinite number of identical interacting systems undergoing deterministic and stochastic forcing. Our approach is able to accommodate both equilibrium and non-equilibrium stationary states, thus going beyond the classical approximation of gradient flows. We remark that the existence of equilibrium stationary (Gibbs) states, the gradient structure (in a suitable metric) and the self-adjointness of the Fokker–Planck operator are equivalent. The presence of interaction between the systems leads to McKean–Vlasov evolution equation for the one-particle density, which reduces to the classical Fokker–Planck equation if the coupling is switched off.

We find explicit expressions for the linear susceptibility and are able to evaluate its asymptotic behaviour, thus allowing for the derivation of a general set of Kramers–Kronig relations and related sum rules. The susceptibility, in close parallel to the classic Clausius–Mossotti expression of macroscopic electric susceptibility for condensed matter, is written in a renormalized form as the product of a matrix describing the self-action of the system times the mean field susceptibility. This allows for further clarifying the relationship between endogenous and exogenous processes, which generalized the fluctuation–dissipation theorem for this class of systems.

Linear response breaks down when the susceptibility diverges, i.e. it develops poles in the real axis. We separate two scenarios of criticality—one associated with the divergence of the mean field susceptibility, and another one associated with singularities of the matrix describing the self-action of the system. The first case pertains to the classical theory of critical transitions.

The second case is here for us of greater interest and can be realized *only* in the thermodynamic limit. We interpret such a second scenario as describing phase transitions for the system. We

define two scenarios of phase transition—a static one, and a dynamic one, where a pole at vanishing frequency and two poles at opposite frequency appear in the linear susceptibility, respectively. At the phase transition, the Kramers–Kronig relations and sum rules valid in the smooth regime of response break down and a detailed study of the poles allows one to find the correction terms. Again, one can establish a link with results from condensed matter physics, as the correction terms resemble those appearing when studying frequency-dependent optical properties of a material at the insulator–metal phase transition, where the static conductivity becomes non-vanishing. We prove that, against intuition, a phase transition is—as opposed to the case of critical transitions—*not* accompanied by a divergence in the autocorrelation properties of the system, i.e. no critical slowing down is observed. Our interpretation is supported by the use of the formalism developed in this paper to revisit through analytical and numerical tools the classical results for phase transitions occurring in the Desai–Zwanzig model on the Bonilla–Casado–Morrillo model, for which it is easy to define appropriate order parameters. The criticalities in the these two models conform to the scenario of static and dynamic phase transition, respectively.

We remark that studying the linear response of the order parameter is the optimal choice for detecting the phase transition but not the only one. In fact, we expect that a broader class of observables can be used in order to identify the critical behaviour. This is especially important in non-equilibrium cases, where the identification of such order parameter can be extremely non-trivial.

The work reported in this paper opens up several avenues for future research. Four natural next steps are: (a) to investigate in greater detail multi-dimensional reversible (equilibrium) McKean–Vlasov dynamics exhibiting phase transitions; for such systems the self-adjointness of the linearized McKean–Vlasov operator enables the systematic use of tools from spectral theory for selfadjoint operators in appropriate Hilbert spaces. (b) To use the analytical tools developed in this paper to design early warning signals for phase transitions, as opposed to critical transitions for which there exists an extensive literature. (c) To better define the class of observables for which the divergence of the linear response can be used to define and detect phase transitions. In particular, we aim at developing systematic analytical and data-driven methodologies for identifying order parameters in agent based models. These tools will enable us to move beyond the quadratic interaction between subsystems that was considered in this paper. (d) To use the framework developed in this paper in order to revisit phenomena such as synchronization, cooperation and consensus in multi-agent systems, and more generally the emergence of coherent structures in complex systems, both in natural and social sciences as well as technology.

Data accessibility. The codes used to run the simulations and the data used for producing the figures are available on figshare.com at https://figshare.com/projects/Response_theory_and_phase_transition_for_thermodynamic_limit_of_interacting_identical_systems/89516.

Authors' contributions. G.P. initiated the study by formulating the general problem and highlighting the relevance of the Desai–Zwanzig and Bonilla–Casado–Morillo models; V.L. developed most of the theoretical framework; N.Z. investigated the Desai–Zwanzig and Bonilla–Casado–Morillo models, performed the numerical simulations and the related data analysis. All authors contributed to the writing of the paper. All authors gave final approval for publication and agree to be held accountable for the work performed therein.

Competing interests.  We declare we have no competing interests.

Funding. V.L. acknowledges the support received by the European Union's Horizon 2020 research and innovation program through the project TiPES (Grant Agreement No. 820970). The work of G.P. was partially funded by the EPSRC, grant no. EP/P031587/1, and by J.P. Morgan Chase & Co. Any views or opinions expressed herein are solely those of the authors listed, and may differ from the views and opinions expressed by J.P. Morgan Chase & Co. or its affiliates. This material is not a product of the Research Department of J.P. Morgan Securities LLC. This material does not constitute a solicitation or offer in any jurisdiction. N.Z. has been supported by an EPSRC studentship as part of the Centre for Doctoral Training in Mathematics of Planet Earth (grant no. EP/L016613/1).

Acknowledgements. V.L. wishes to thank A. Pikovsky for his very insightful criticism on the applicability of response theory near the regime of synchronization; and D. Sornette for some useful exchanges on exogenous versus endogeous dynamics in the course of a virtual conference.

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
