## [Reviewer comments · Proceedings. Mathematical, Physical, and Engineering Sciences]

Review History

RSPA-2020-0688.R0 (Original submission)

Review form: Referee 1

Is the manuscript an original and important contribution to its field?

Acceptable

Is the paper of sufficient general interest?

Acceptable

Is the overall quality of the paper suitable?

Acceptable

Do you have any ethical concerns with this paper?

No

Recommendation?

Major revision is needed (please make suggestions in comments)

Comments to the Author(s)

The paper deals with aspects of response theory in the case of phase transitions. The paper appears rather technical and not very readable. The paper would probably benefit from greater focus. E.g. there is an introduction (not so short) which goes in many directions but that does not

really addresses the new result of the paper. What do we learn here? Apart from the general concern about focus and readability (what is truly the contribution of the paper?) there are some questions that perhaps the authors are willing to clarify:

1. is it correct that in all cases of the analysis we deal with harmonic interaction between the particles? If yes and if important for the results, why not say it clearer and sooner? What physics, sociology, meteorology or climate is the model representing, even as a toy? What is the nature of the non-equilibrium that the authors want to study, if at all? Readers would perhaps be happy to concentrate on the Bonilla-Casado-Morilla model and learn what is new and relevant there? What is the best example or illustration we get from the paper?
2. We learn in school that the Kramers-Kronig relations are generally valid under some conditions. Can we not use that directly from the standard text books? Is there something unique here?
3. Are the linear response formula constructive? Why these observables and perturbations? Are those important for an applicaton? And what do the resulting formulae give in the end? Is the paper 'proving validity of linear response?'

These are examples fo questions that illustrate the difficulty for the reader. The paper and the interesting subject would benefit from greater focus and clarity on the things that truly matter and on discussion on what specifically is gained from the paper.

Review form: Referee 2

Is the manuscript an original and important contribution to its field?

Good

Is the paper of sufficient general interest?

Acceptable

Is the overall quality of the paper suitable?

Excellent

Can the paper be shortened without overall detriment to the main message?

Yes

Do you think some of the material would be more appropriate as an electronic appendix?

No

Do you have any ethical concerns with this paper?

No

Recommendation?

Accept as is

Comments to the Author(s)

The manuscript is very clearly written and discusses in details an important approach to a class of phenomena of great interest. Thus, I recommend its publication as it is.

Decision letter (RSPA-2020-0688.R0)

27-Oct-2020

Dear Dr Lucarini

The Editor of Proceedings A has now received comments from referees on the above paper and would like you to revise it in accordance with their suggestions which can be found below (not including confidential reports to the Editor).

Please submit a copy of your revised paper within four weeks - if we do not hear from you within this time then it will be assumed that the paper has been withdrawn. In exceptional circumstances, extensions may be possible if agreed with the Editorial Office in advance.

Please note that it is the editorial policy of Proceedings A to offer authors one round of revision in which to address changes requested by referees. If the revisions are not considered satisfactory by the Editor, then the paper will be rejected, and not considered further for publication by the journal. In the event that the author chooses not to address a referee's comments, and no scientific justification is included in their cover letter for this omission, it is at the discretion of the Editor whether to continue considering the manuscript.

- Acknowledgements
- Funding statement

To revise your manuscript, log into <http://mc.manuscriptcentral.com/prsa> and enter your Author Centre, where you will find your manuscript title listed under "Manuscripts with Decisions." Under "Actions," click on "Create a Revision." Your manuscript number has been appended to denote a revision.

You will be unable to make your revisions on the originally submitted version of the manuscript. Instead, revise your manuscript and upload a new version through your Author Centre.

When submitting your revised manuscript, you will be able to respond to the comments made by the referee(s) and upload a file "Response to Referees" in "Section 6 - File Upload". Please use this to document how you have responded to the comments, and the adjustments you have made. In order to expedite the processing of the revised manuscript, please be as specific as possible in your response to the referee(s).

IMPORTANT: Your original files are available to you when you upload your revised manuscript. Please delete any unnecessary previous files before uploading your revised version.

When revising your paper please ensure that it remains under 28 pages long. In addition, any pages over 20 will be subject to a charge (£150 + VAT (where applicable) per page). Your paper has been ESTIMATED to be 25 pages.

Once again, thank you for submitting your manuscript to Proc. R. Soc. A and I look forward to receiving your revision. If you have any questions at all, please do not hesitate to get in touch.

Yours sincerely
Raminder Shergill
proceedingsa@royalsociety.org

on behalf of
Dr Marco Mazza
Board Member
Proceedings A

Reviewer(s)' Comments to Author:

Referee: 1

Comments to the Author(s)

The paper deals with aspects of response theory in the case of phase transitions. The paper appears rather technical and not very readable. The paper would probably benefit from greater focus. E.g. there is an introduction (not so short) which goes in many directions but that does not really address the new result of the paper. What do we learn here? Apart from the general concern about focus and readability (what is truly the contribution of the paper?) there are some questions that perhaps the authors are willing to clarify:

1. Is it correct that in all cases of the analysis we deal with harmonic interaction between the particles? If yes and if important for the results, why not say it clearer and sooner? What physics, sociology, meteorology or climate is the model representing, even as a toy? What is the nature of the non-equilibrium that the authors want to study, if at all? Readers would perhaps be happy to concentrate on the Bonilla-Casado-Morilla model and learn what is new and relevant there? What is the best example or illustration we get from the paper?
2. We learn in school that the Kramers-Kronig relations are generally valid under some conditions. Can we not use that directly from the standard text books? Is there something unique here?
3. Are the linear response formulae constructive? Why these observables and perturbations? Are those important for an application? And what do the resulting formulae give in the end? Is the paper "proving validity of linear response?"

These are examples of questions that illustrate the difficulty for the reader. The paper and the interesting subject would benefit from greater focus and clarity on the things that truly matter and on discussion on what specifically is gained from the paper.

Referee: 2

Comments to the Author(s)

The manuscript is very clearly written and discusses in details an important approach to a class of phenomena of great interest. Thus, I recommend its publication as it is.

RSPA-2020-0688.R1 (Revision)

Review form: Referee 1

Is the manuscript an original and important contribution to its field?

Acceptable

Is the paper of sufficient general interest?

Good

Is the overall quality of the paper suitable?

Good

Can the paper be shortened without overall detriment to the main message?

Yes

Do you have any ethical concerns with this paper?

No

Recommendation?

Accept as is

Comments to the Author(s)

The authors made a considerable effort to come a good way in answering to the previous comments. That is much appreciated and believed to make the paper ready now for publication. Warm recommendation.

Decision letter (RSPA-2020-0688.R1)

25-Nov-2020

Dear Dr Lucarini

I am pleased to inform you that your manuscript entitled "Response Theory and Phase Transitions for the Thermodynamic Limit of Interacting Identical Systems" has been accepted in its final form for publication in Proceedings A.

Our Production Office will be in contact with you in due course. You can expect to receive a proof of your article soon. Please contact the office to let us know if you are likely to be away from e-mail in the near future. If you do not notify us and comments are not received within 5 days of sending the proof, we may publish the paper as it stands.

Open access

You are invited to opt for open access, our author pays publishing model. Payment of open access fees will enable your article to be made freely available via the Royal Society website as soon as it is ready for publication. For more information about open access please visit <https://royalsociety.org/journals/authors/which-journal/open-access/>. The open access fee for this journal is £1700/\$2380/€2040 per article. VAT will be charged where applicable.

Note that if you have opted for open access then payment will be required before the article is published – payment instructions will follow shortly.

If you wish to opt for open access then please inform the editorial office (proceedingsa@royalsociety.org) as soon as possible.

Your article has been estimated as being 27 pages long. Our Production Office will inform you of the exact length at the proof stage.

Proceedings A levies charges for articles which exceed 20 printed pages. (based upon approximately 540 words or 2 figures per page). Articles exceeding this limit will incur page charges of £150 per page or part page, plus VAT (where applicable).

Under the terms of our licence to publish you may post the author generated postprint (ie. your accepted version not the final typeset version) of your manuscript at any time and this can be made freely available. Postprints can be deposited on a personal or institutional website, or a recognised server/repository. Please note however, that the reporting of postprints is subject to a media embargo, and that the status the manuscript should be made clear. Upon publication of the definitive version on the publisher's site, full details and a link should be added.

You can cite the article in advance of publication using its DOI. The DOI will take the form: 10.1098/rspa.XXXX.YYYY, where XXXX and YYYY are the last 8 digits of your manuscript

number (eg. if your manuscript number is RSPA-2017-1234 the DOI would be 10.1098/rspa.2017.1234).

For tips on promoting your accepted paper see our blog post:
<https://royalsociety.org/blog/2020/07/promoting-your-latest-paper-and-tracking-your-results/>

On behalf of the Editor of Proceedings A, we look forward to your continued contributions to the Journal.

Sincerely,
Raminder Shergill
proceedingsa@royalsociety.org

Reviewer(s)' Comments to Author:

Referee: 1

Comments to the Author(s)

The authors made a considerable effort to come a good way in answering to the previous comments. That is much appreciated and believed to make the paper ready now for publication.
Warm recommendation.